# Brassinosteroid signaling delimits root gravitropism via sorting of the *Arabidopsis* PIN2 auxin transporter

Katarzyna Retzer [1,2,6], Maria Akhmanova[3,6], Nataliia Konstantinova[1], Kateřina Malínská [2],
Johannes Leitner[1,5], Jan Petrášek[2,4] & Christian Luschnig [1*]

*Arabidopsis* PIN2 protein directs transport of the phytohormone auxin from the root tip into the root elongation zone. Variation in hormone transport, which depends on a delicate interplay between PIN2 sorting to and from polar plasma membrane domains, determines root growth. By employing a constitutively degraded version of PIN2, we identify brassino-lides as antagonists of PIN2 endocytosis. This response does not require de novo protein synthesis, but involves early events in canonical brassinolide signaling. Brassinolide-controlled adjustments in PIN2 sorting and intracellular distribution governs formation of a lateral PIN2 gradient in gravistimulated roots, coinciding with adjustments in auxin signaling and directional root growth. Strikingly, simulations indicate that PIN2 gradient formation is no prerequisite for root bending but rather dampens asymmetric auxin flow and signaling. Crosstalk between brassinolide signaling and endocytic PIN2 sorting, thus, appears essential for determining the rate of gravity-induced root curvature via attenuation of differential cell elongation.

[1] Department of Applied Genetics and Cell Biology, University of Natural Resources and Life Sciences, Vienna (BOKU), Muthgasse 18, 1190 Wien, Austria.
[2] Institute of Experimental Botany of the Czech Academy of Sciences, Rozvojová 263 Praha 6, 165 02 Praha, Czech Republic. [3] Institute of Science and Technology Austria (IST Austria), Am Campus 1, 3400 Klosterneuburg, Austria. [4] Department of Experimental Plant Biology, Faculty of Science, Charles University, 128 44Prague 2, Prague, Czech Republic. [5] Present address: Boehringer Ingelheim RCV, Doktor-Boehringer-Gasse 5-11, 1120 Wien, Austria. [6]These authors contributed equally: Katarzyna Retzer, Maria Akhmanova. *email: christian.luschnig@boku.ac.at

D irectional transport of the growth regulator auxin throughout the plant body establishes morphogenetic cues, influencing a range of developmental programs. PIN-FORMED (PIN) proteins mediate such polar auxin transport (PAT) by means of their asymmetric distribution at the plasma membrane (PM) and are subject to tight control, influencing localization, activity, and abundance[1–3].

PIN exocytotic sorting to distinct polar PM domains[1,2,4,5] is followed by endocytic sorting via clathrin-mediated endocytosis (CME) into trans-Golgi network (TGN) compartments[6]. Such cargo is either recycled to PM domains, or passed on to Late Endosomes (LE)/Multivesicular Bodies (MVB) for degradation in the vacuole[5,7]. Cis- and trans-acting regulators, determining the fate of endocytosed PINs, have been identified. ARF-GEFs and components of a plant retromer complex, for example, mediate rerouting to the PM, whereas protein ubiquitylation triggers Endosomal sorting complex required for transport (ESCRT)-dependent PIN vacuolar targeting[5,7–10]. Consequently, variations in PIN abundance and subcellular distribution define directional, intercellular auxin flow, instrumental for establishment of auxin gradients, thus shaping plant growth and rapid adaptation in response to fluctuating environmental conditions.

Plant hormones impact on PIN sorting and abundance. Auxin for example has been suggested to stabilize PIN proteins at the PM, but to induce vacuolar sorting and degradation after prolonged incubation[11,12]. Stabilizing effects have been attributed to the activity of gibberellic acid (GA)[13,14], whereas differing concentrations of jasmonate promote either stabilization or degradation of PIN2[15]. Cytokinin and the strigolactone agonist rac-GR24 promote PIN endocytic sorting and vacuolar degradation, thereby influencing particular aspects of plant morphogenesis[16,17]. Furthermore, secondary messengers, such as calcium and redox signaling affect PIN sorting during adaptation processes, emphasizing the central role of crosstalk between plant growth regulators and PIN proteins in plant development[18,19].

Brassinosteroids represent plant steroid hormones, regulating cell proliferation and differentiation, often acting in conjunction with additional hormones[20–24]. Crosstalk between brassinolide and auxin signaling shapes plant development, predominantly via transcriptional control[25–27]. Additionally, brassinolide signaling has been implicated in sorting and steady-state level control of root-specific PIN2[1,28–30]; however, mechanisms orchestrating such post-transcriptional control and its significance for auxin transport remained obscure.

Here, by employing a constitutively endocytosed allele of PIN2, we demonstrate that brassinolide modulates vacuolar degradation of PIN2, specifically in gravity-responding roots. Assessment of crosstalk between brassinolide and PIN2 sorting, together with modeling of auxin flow in gravistimulated roots, highlight a mechanism, by which brassinolide delimits root curvature via differential sorting of PIN2.

## Results

**Brassinolide controls turnover of ubiquitylated PIN2.** Sorting of PIN2 is modulated by various stimuli, defining auxin flow and, consequently, root growth in response to them. To extend our knowledge about PIN2 regulation, we tested for effects on the sorting and abundance of the ubiquitin-tagged PIN2:ubq:VEN[7]. This reporter protein mimics constitutive PIN2 ubiquitylation, resulting in its enhanced internalization and vacuolar targeting in dependence of the ESCRT endocytic sorting machinery[7,8] (Supplementary Fig. 1a, b). We crossed the reporter line into tamoxifen-inducible pINTAM»RFP:HUB, exerting dominant negative effects on CME from the PM[6], revealing retention of PIN2:ubq:VEN signals at the PM, specifically in cells expressing

RFP:HUB (Supplementary Fig. 1c–i). Thus, similar to sorting of ubiquitylated PM cargo in fungi and metazoa, endocytic sorting of PIN2:ubq:VEN in Arabidopsis involves activities of CME and the ESCRT machinery[31].

When testing for signals affecting endocytosis of ubiquitylated PIN2, we observed strong responses to the plant steroid hormone 24-epibrassinolide (eBL). This growth regulator caused a concentration- and time-dependent quantitative increase in PIN2:ubq:VEN abundance on Western blots and upon assessment of fluorescent signal intensities (Fig. 1a–h), with low eBL concentrations (0.1 nM) resulting in increased reporter protein signals after extended incubation times (Supplementary Fig. 2a–c).

Detailed examination of eir1-4 PIN2p::PIN2:ubq:VEN root epidermis cells demonstrated accumulation of reporter signals predominantly at the PM in response to eBL (Fig. 1a–d). These effects appear specific, since treatment with another active brassinosteroid, 28-homobrassinolide, also caused stabilization of PIN2:ubq:VEN, whereas no comparable responses were observed when testing additional steroid compounds (Fig. 1f, i, Supplementary Fig. 3d).

We then determined, whether upregulation of PIN2:ubq:VEN could result from eBL effects on transcription. However, neither PIN2:ubq:VEN nor endogenous PIN2 transcript levels increased in response to eBL (Fig. 1j). Furthermore, analysis of PIN2 expressed under control of the strong RP40 ribosomal protein promoter[32] showed that unlike RP40p::PIN2:VEN roots, which displaying prominent reporter signals at the PM of stele, ground tissue and epidermis cells, only faint intracellular signals could be observed in RP40p::PIN2:ubq:VEN roots (Supplementary Fig. 3a, b). Thus, constitutive ubiquitylation induces internalization and degradation of PIN2, regardless of ectopic expression. Furthermore, treatment of RP40p::PIN2:ubq:VEN with eBL, caused reporter protein stabilization at the PM, recapitulating results obtained with PIN2p::PIN2:ubq:VEN (Fig. 1k, l; Supplementary Fig. 3c, d).

**Brassinolide homeostasis and BRI1 control PIN2:ubq:VEN fate.** To identify pathways connecting brassinolide and ubiquitylated PIN2, we tested det2-1 eir1-4 PIN2p::PIN2:ubq:VEN, which is affected in DEETIOLATED2, required for an early step in brassinosteroid biosynthesis[33]. PIN2:ubq:VEN reporter signals were even weaker than in eir1-4 PIN2p::PIN2:ubq:VEN, suggesting that interference with brassinolide biosynthesis results in further destabilization of ubiquitylated PIN2 (Fig. 2a, b, d). Consistently, eBL application, which rescues det2-1[33], restored reporter expression in det2-1 eir1-4 PIN2p::PIN2:ubq:VEN (Fig. 2c, d). In agreement, treatment of eir1-4 PIN2p::PIN2:ubq:VEN with Brassinazole (BRZ), a potent inhibitor of brassinolide biosynthesis[34], resulted in reduced abundance of PIN2:ubq:VEN at the PM (Fig. 2e–g), which underlines a role for brassinolide homeostasis in regulating the stability of ubiquitylated PIN2.

Brassinolide is a positive regulator of GA biosynthesis[35], another phytohormone modulating PIN sorting[13,14]. Thus, we tested for involvement of eBL-controlled GA homeostasis in the regulation of PIN2:ubq:VEN. Co-treatment with eBL and paclobutrazole (PAC), an inhibitor of GA biosynthesis, did not affect eBL-induced PIN2:ubq:VEN localization at the PM, which argues for brassinolide effects uncoupled from GA biosynthesis (Supplementary Fig. 4a–e).

Next, to characterize the signaling pathways involved, we generated a cross between a mutant in BRASSINOSTEROIDE INSENSITIVE1 brassinolide receptor kinase (bri1-6) and eir1-4 PIN2p::PIN2:ubq:VEN. Unlike in eir1-4 PIN2p::PIN2:ubq:VEN, eBL treatment of bri1-6 eir1-4 PIN2p::PIN2:ubq:VEN did not efficiently restore PIN2:ubq:VEN abundance at the PM,

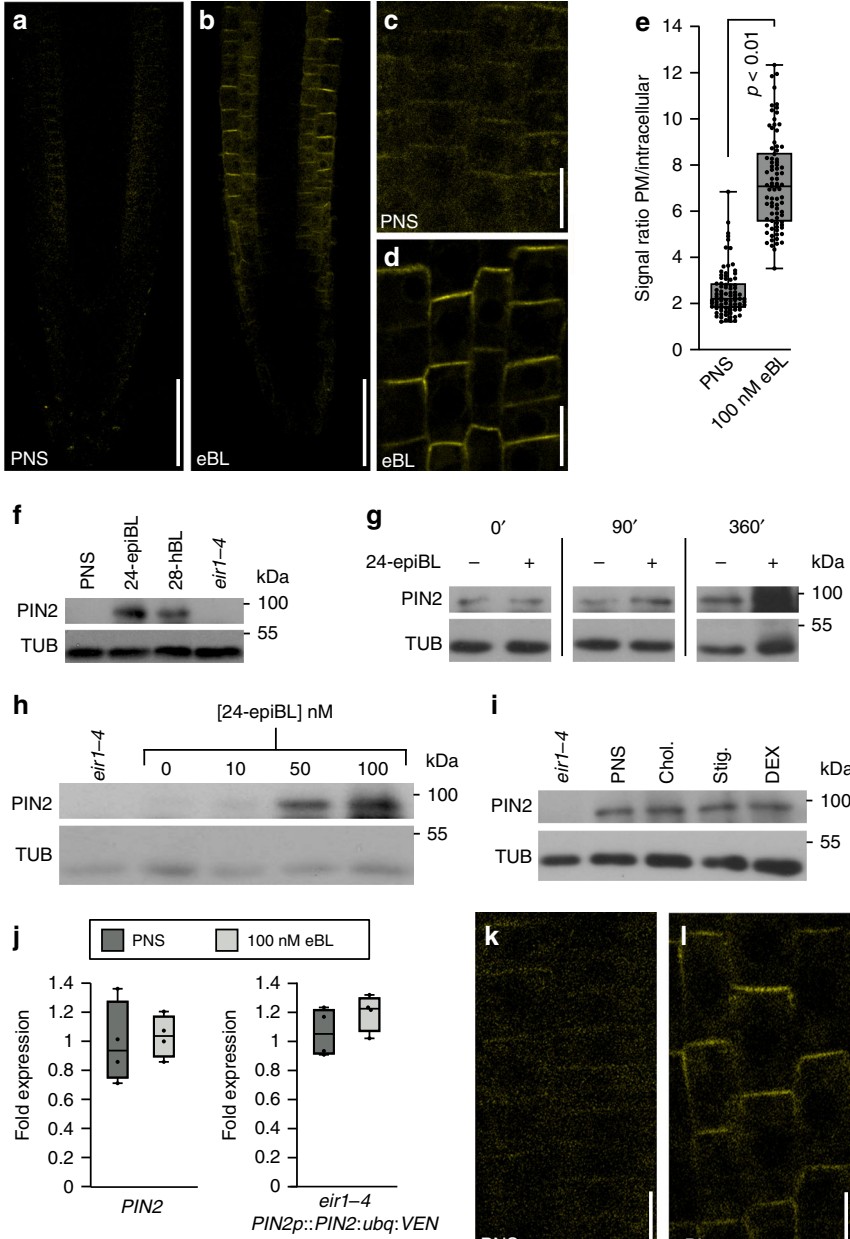

**Fig. 1 Brassinolide stabilizes PIN2:ubq:VEN. a–d** Comparison of *eir1-4 PIN2p::PIN2:ubq:VEN* root tips (**a**, **b**) and root epidermis cells (**c**, **d**) at 6 DAG germinated on PNS (**a**, **c**) or on PNS supplemented with 100 nM eBL (**b**, **d**). **e** PIN2:ubq:VEN signal ratios between PM and the intracellular space in PNS and eBL-treated seedlings. 82–85 root epidermal cells in 12 roots were tested for each dataset and analyzed by two-tailed *t*-test. **f–i** Western blots performed with *eir1-4 PIN2p::PIN2:ubq:VEN* membrane protein fractions isolated at 6 DAG probed with anti-PIN2; where indicated *eir1-4* was used as a control. Anti-α-tubulin (TUB) was used as a loading control for all blots. **f** Samples either remained mock-treated (PNS) or were germinated in the presence of 100 nM 24-epiBL, or 28-homoBL. **g** Samples taken from a time course after 0, 90, and 360 min in the presence of 100 nM eBL ( + ) or incubated with solvent alone (-). **h** Samples with increasing concentrations of 24-epiBL. **i** Samples treated with 50 µM cholesterol (Chol.), 50 µM stigmasterol (Stig.), and 50 µM dexamethasone (DEX) for 16 h, or mock-treated sample (PNS). **j** *PIN2* transcript levels in 6-day-old Col0 and *PIN2p::PIN2:ubq:VEN* on PNS or on 100 nM eBL for 6 h. Four biological repetitions were made for each sample, with transcripts normalized to expression of *EF1a* (At1g07940). **k, l** PIN2:ubq: VEN distribution and abundance in root epidermis cells of *RP40p::PIN2:ubq:VEN* at 6 DAG, on PNS (**k**) or in the presence of 100 nM eBL for 6 h (**l**). Whiskers in box plots cover the entire range of outliers obtained in the datasets; gray boxes: first and third quartiles; center line: median; dots: values obtained. Scale bars: a,b = 50 µm; c,d = 20 µm; k,l = 10 µm. Source data are provided as Source Data file.

suggesting involvement of *BRI1* in regulating the trafficking of the ubiquitylated PIN2 (Fig. 2h–j, l–o). Bikinin is a potent inhibitor of GSK3/Shaggy-type serine/threonine kinases, which function as negative regulators of brassinosteroid signaling downstream of *BRI1*[36,37]. Inhibition of these protein kinases therefore causes activation of brassinolide signaling, even in *bri1* loss-of-function

mutants[38]. Bikinin treatment stabilized PIN2:ubq:VEN at the PM in *eir1-4 PIN2p::PIN2:ubq:VEN* and in *bri1-6 eir1-4 PIN2p::PIN2: ubq:VEN* roots (Fig. 2h, k, n, o). Overall, this demonstrates involvement of canonical brassinolide signaling elements, namely *BRI1* and GSK3/Shaggy-type kinases, in PIN2:ubq:VEN trafficking.

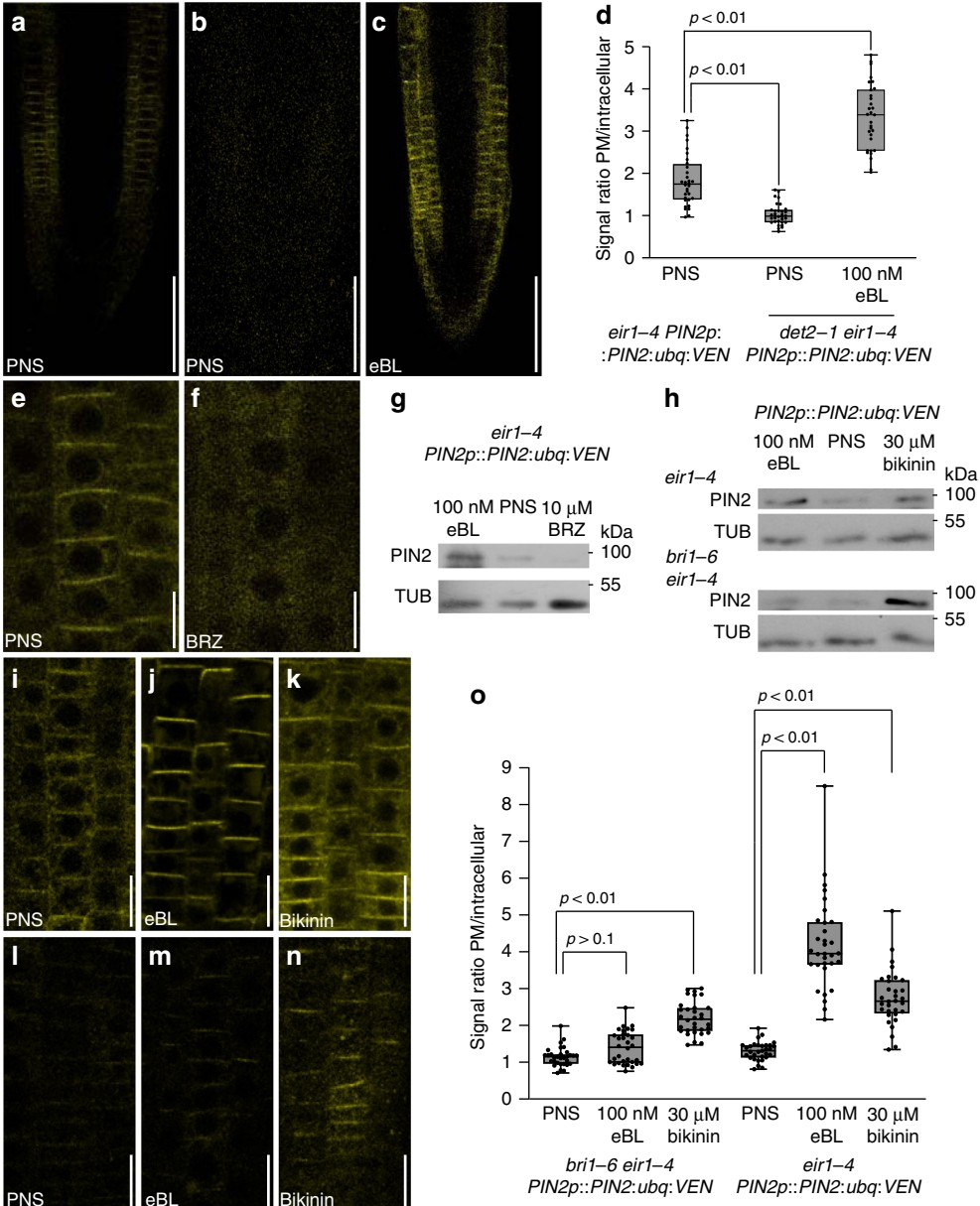

**Fig. 2 PIN2:ubq:VEN is controlled by brassinolide homeostasis and canonical eBL signaling. a–c** PIN2:ubq:VEN in root tips of *eir1-4 PIN2p::PIN2:ubq:VEN* (**a**) and *det2-1 eir1-4 PIN2p::PIN2:ubq:VEN* grown in the absence (**b**) or presence of 100 nM eBL at 6 DAG (**c**). **d** PIN2:ubq:VEN signal ratios between PM and intracellular space in *eir1-4 PIN2p::PIN2:ubq:VEN* and *det2-1 eir1-4 PIN2p::PIN2:ubq:VEN*. 31-32 root epidermis cells in five roots were tested for each dataset. Relevant *p*-values obtained by One-way ANOVA with post-hoc Tukey HSD test are indicated. **e, f** PIN2:ubq:VEN distribution and abundance in 6 DAG *eir1-4 PIN2p::PIN2:ubq:VEN* root epidermis cells incubated on PNS (**e**) or after 16 h incubation on 10 µM Brassinazole (BRZ; **f**). **g** Western blots of *eir1-4 PIN2p::PIN2:ubq:VEN* membrane protein fraction, probed with anti-PIN2. Samples were incubated for 16 h in the presence of 100 nM 24-epiBL, 10 µM BRZ, or solvents only (PNS). **h** Western blots of *eir1-4 PIN2p::PIN2:ubq:VEN* (top panel) and *bri1-6 eir1-4 PIN2p::PIN2:ubq:VEN* (bottom panel) membrane protein fraction, probed with anti-PIN2. Samples were incubated for 16 h in the presence of 100 nM 24-epiBL, 30 µM bikinin, or on solvents only (PNS). **i–n** PIN2:ubq:VEN in root tips of *eir1-4 PIN2p::PIN2:ubq:VEN* (**i–k**) and *bri1-6 eir1-4 PIN2p::PIN2:ubq:VEN* (**l–n**) on PNS (**i**, **l**), or after incubation on 100 nM eBL (**j**, **m**) or 30 µM bikinin (**k**, **n**) for 16 h. **o** PIN2:ubq:VEN signal ratios between PM and intracellular space in *eir1-4 PIN2p::PIN2:ubq:VEN* and *bri1-6 eir1-4 PIN2p::PIN2:ubq:VEN*. 31-32 root meristem epidermis cells in 6 roots were tested for each dataset. Relevant p-values obtained by One-way ANOVA with post-hoc Tukey HSD test are indicated. Anti-α-tubulin (TUB) was used as loading control for Western blots. Whiskers in box plots cover the entire range of outliers obtained in the datasets; gray boxes: first and third quartiles; center line: median; dots: values obtained. Scale bars: a–c = 50 µm; e,f,i–n = 10 µm. Source data are provided as Source Data file.

**Brassinolide antagonizes PIN2:ubq:VEN endocytic sorting**. We asked, whether stabilization of PIN2:ubq:VEN results from retention at, or from enhanced sorting to the PM. We tested this in *eir1-4 PIN2p::PIN2:ubq:VEN* seedlings by cycloheximide treatment (CHX), to block de novo protein biosynthesis, followed by treatment with CHX and Brefeldin A (BFA), an inhibitor of anterograde PIN trafficking[5]. This resulted in intracellular accumulation of Venus signals, indicating reporter protein retention in response to BFA (Fig. 3a, d). Pre-treatment with eBL did not interfere with this process, whilst after BFA washout, intracellular

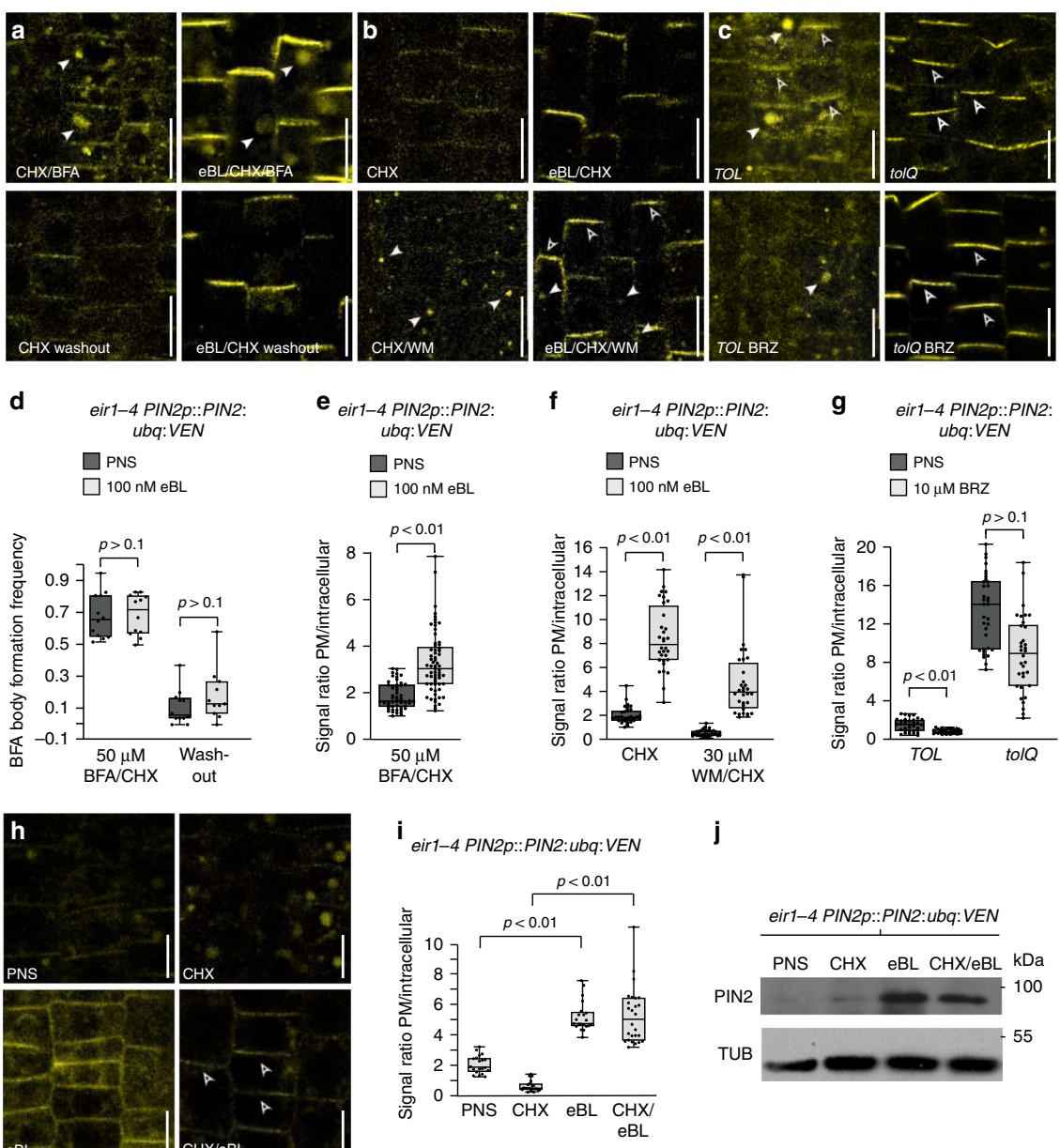

**Fig. 3 Brassinolide antagonizes endocytic sorting of constitutively ubiquitylated PIN2. a** 6 DAG *eir1-4 PIN2p::PIN2:ubq:VEN* on PNS, or treated with 100 nM eBL for 16 h, followed by 50 μM CHX or 100 nM eBL/50 μM CHX treatment for 30′, and by co-incubation for 60′ in the presence of 50 μM CHX/ 50 μM BFA and 100 nM eBL/50 μM CHX/50 μM BFA. White arrowheads: BFA-induced compartments. **b** 6-day-old *eir1-4 PIN2p::PIN2:ubq:VEN* grown on PNS or treated with 100 nM eBL for 16 h, followed by treatment with 50 μM CHX for 30′ (CHX), or co-treated with eBL and CHX for 30′ (eBL/CHX). This was followed by co-incubation in the presence of either 50 μM CHX and 30 μM WM (CHX/WM) or 100 nM eBL, 50 μM CHX and 30 μM WM (eBL/ CHX/WM) for another 90′. White arrowheads: PIN2:ubq:VEN signals in WM-induced endocytic compartments; open arrowheads: PM localization. **c** PIN2: ubq:VEN signals in 6 DAG *TOL* and *tolQ* root epidermis cells, on PNS or on 10 μM BRZ for 16 h. White arrowheads: intracellular signals; open arrowheads: PM localization. **d** Frequency of BFA-compartment formation observed in 'a' (n = 10–12 roots and 230–550 cells/dataset). **e** Signal ratios for 'a' (n = 8 roots and 52/65 trichoblast cells/dataset). **f** Signal ratios for 'b' (n = 5 roots and 32–33 cells/dataset). **g** Signal ratios for 'c' (n = 4 roots and 34 cells/dataset). **h** Root meristem cells *eir1-4 PIN2p::PIN2:ubq:VEN* seedlings, on PNS (left, top); on 50 μM CHX for 5.5 h (top, right); on 100 nM eBL for 5 h (bottom, left); on 50 μM CHX for 30′, followed by eBL/CHX for 5 h (bottom, right). Open arrowheads: signals at the PM. **i** Signal ratios for 'h' (n = 4 roots and 20–26 cells/ dataset). **j** *eir1-4 PIN2p::PIN2:ubq:VEN* root membrane fractions, probed for PIN2 and TUB (α-tubulin). Conditions as in 'h'. Box plot whiskers represent the entire range of outliers; gray boxes: first and third quartiles; center line: median; dots: values obtained. Two-tailed *t*-test analysis has been performed to test for *p*-values. Scale bars: a–c, h = 10 μm. Source data are provided as a Source Data file.

aggregates vanished in eBL-treated and in control samples (Fig. 3a, d). Together, this indicates that PIN2:ubq:VEN retention in BFA compartments occurs regardless of eBL treatment. However, quantification of PIN2:ubq:VEN signal intensities at the PM vs. BFA compartments, indicated that a smaller proportion of

intracellular reporter signals accumulated in eBL-treated samples, potentially reflecting reduced endocytic sorting of PIN2:ubq:VEN in response to eBL (Fig. 3a, e).

Treatment of *eir1-4 PIN2p::PIN2:ubq:VEN* with Wortmannin (WM), a phosphatidylinositol kinase inhibitor that obstructs

vacuolar cargo sorting[39], caused formation of punctate intracellular Venus signals, indicating its ongoing vacuolar targeting (Fig. 3b). Pre-treatment with eBL followed by WM treatment, did not completely block PIN2:ubq:VEN signal accumulation, but caused reduced internalization of it (Fig. 3b, f), signifying inhibition of PIN2:ubq:VEN endocytic sorting in response to brassinolide. Consistent results were obtained when testing BRZ effects in an *Arabidopsis TOM1-LIKE* (*TOL*) pentuple mutant (*tolQ*). This mutant is compromised in an early recognition step of ubiquitylated membrane protein cargo, causing PIN2:ubq:VEN retention at the PM, instead of its endocytic sorting[8]. Unlike wild type, in which BRZ-induced inhibition of brassinolide biosynthesis enhances vacuolar targeting of PIN2:ubq:VEN, *tolQ* antagonizes these BRZ effects (Fig. 3c, g). Variations in PIN2:ubq:VEN endocytosis caused by altered brassinolide homeostasis thus, require the TOL/ESCRT machinery, guiding ubiquitylated membrane proteins from the PM towards the vacuolar compartment.

Next, we asked if de novo protein biosynthesis is required for eBL effects observed. Inhibition of protein biosynthesis by CHX under our conditions was first validated by analysis of the instable D2-Venus reporter protein[40] (Supplementary Fig. 5a–d). We then treated *eir1-4 PIN2p::PIN2:ubq:VEN* with CHX for 30 min, followed by co-incubation in the presence of CHX and eBL for 5 h and observed PIN2:ubq:VEN signals at the PM, which were absent in untreated controls or in the presence of CHX alone (Fig. 3h, i). Consistently, Western analysis of membrane protein extracts showed accumulation of PIN2:ubq:VEN after pre-treatment with CHX, followed by CHX/eBL co-treatment; a response not observed in controls (Fig. 3j). Pre-treatment of *eir1-4 PIN2p::PIN2:ubq:VEN* with eBL, followed by translational inhibition for 150 min, led to PIN2:ubq:VEN PM retention as well (Supplementary Fig. 6a–e), corroborating that de novo protein biosynthesis is dispensable for brassinolide effects on PIN2:ubq:VEN.

**Brassinolide modulates endocytosis of wild type PIN2.** We then determined brassinolide responsiveness of Venus-tagged wild type PIN2 (PIN2:VEN). BRZ-mediated inhibition of brassinolide biosynthesis in *eir1-4 PIN2p::PIN2:VEN* caused moderate increases in intracellular Venus signals (Supplementary Fig. 7a, b). In line with these observations, *bri1-6 PIN2p::PIN2:VEN* root meristem epidermis cells exhibited punctate intracellular reporter signals, significantly more abundant than in *BRI1* controls (Fig. 4a–c; Supplementary Fig. 7c, d). Co-staining with the endocytosed marker FM4-64, demonstrated overlaps in the distribution of internalized FM4-64 and PIN2:VEN in *bri1-6* root epidermis cells, suggesting that deficiencies in brassinolide perception promote internalization of PIN2 (Fig. 4a–c). Despite alterations in PIN2 distribution, Western blots revealed no striking changes in PIN2 abundance, neither in response to BRZ treatment, nor in *bri1-6* (Supplementary Fig. 7e, f). Furthermore, no striking adjustments in PIN2:VEN PM localization and/or steady-state protein levels were observed in response to eBL (Fig. 4d–f; Supplementary Fig. 7g, h). Thus, alterations in brassinolide homeostasis or signaling seemingly affect PIN2 sorting, without noticeable effects on total protein levels. Related findings were made, when assessing eBL effects on expression of additional auxin transport proteins, which revealed no striking differences in response to brassinolide (Supplementary Fig. 8a–d). Next, we tested conditions, demonstrated to cause vacuolar degradation of PIN2[7,9]. Dark growth of *eir1-4 PIN2p::PIN2:mCherry* roots caused pronounced vacuolar reporter signals in root epidermis cells, reflecting enhanced degradation of the reporter protein. This was reverted by eBL treatment, suggesting that brassinolide antagonizes PIN2 vacuolar targeting (Fig. 4g, h; Supplementary Fig. 7i–l). Next, we incubated *eir1-4 PIN2p::PIN2:*

*VEN* in the presence of auxin analog 1-NAA (1-Naphthaleneacetic acid) for 4-to-5 h, which induced PIN2:VEN internalization, together with a reduction in total protein levels (Fig. 4i, k, l). Auxin-induced PIN2:VEN degradation was antagonized by eBL, reflected in a partial restoration of reporter protein levels and increased PM localization of PIN2:VEN signals, when compared to auxin-treated controls (Fig. 4j–l). Collectively, these findings indicate a role for brassinolide in PIN2 regulation, by countering its vacuolar targeting and degradation.

**Brassinolide acts in PIN2-controlled directional root growth.** Brassinolide influences root growth via distinct mechanisms, including control of cell cycle progression, cytoskeleton configuration and cell wall biosynthesis[22–24,26,29]. PIN2, on the other hand, represents a regulator of directional root growth, mediating auxin flow into the root elongation zone (EZ). To understand how brassinolide could influence root growth via PIN2, we compared phenotypes of wild type and *eir1-4* roots in response to eBL. Moderate hormone concentrations promote curliness of *Arabidopsis* root growth, a response, earlier linked to brassinolide-induced reconfigurations of the cytoskeleton[29]. Indeed, wild type roots germinated on 1 nM eBL responded with formation of small bends and turns, and a similar increase in irregular root twists was observed in *eir1-4* under our growth conditions (Supplementary Fig. 9a). Thus, PIN2 seems dispensable for brassinolide-induced root curling. We then examined gravitropic root bending, as this growth response, which requires PIN2, is mediated by differential cell elongation at the upper vs. the lower side of bending roots[41]. Wild type plants, when germinated on vertically oriented plates in the presence of brassinolide, exhibited deficiencies in directional root growth in a dosage-dependent manner (Fig. 5a, d) In contrast, brassinolide did not cause any prominent changes in already agravitropic *eir1-4* root growth (Fig. 5a, c, e).

We tested eBL effects on the kinetics of root gravitropism, by monitoring reorientation of horizontally positioned primary roots over time. In line with published research, we found that brassinolide treatment does not interfere with root bending in response to gravistimulation[42–44], but quite the opposite, we observed a prominent hyper-responsiveness to gravistimulation (Fig. 5f). Whilst seedlings grown on regular medium terminated root bending, once the direction of root growth has realigned with the direction of the gravity vector, eBL-treated roots had a tendency to over-bend. As a result, eBL-treated roots frequently exhibited a hairpin-shaped root growth pattern, which we did not observe in mock-treated controls (Fig. 5g, h). Gravistimulated *eir1-4 PIN2p::PIN2:VEN* on eBL exhibited growth characteristics similar to wild type and furthermore, a partial rescue of gravitropism defects was observed with eBL-treated *eir1-4 PIN2p::PIN2:ubq:VEN* (Supplementary Fig. 9b). Thus, stabilization of ubiquitylated PIN2 in response to brassinolide appears to contribute to restoration of differential auxin flow and cell elongation. In contrast, *eir1-4* roots treated with eBL exhibited increased irregularity in the directionality of root growth, but we did not observe a comparable rescue of root gravitropism (Fig. 5i; Supplementary Fig. 9b). Related observations were made with the auxin co-receptor quadruple mutant *tir1-1 afb1-1 afb2-1 afb3-1*[45] (*TRANSPORT INHIBITOR RESPONSE1, AUXIN SIGNALING F-BOX*; Supplementary Fig. 9c). Thus, brassinolide action in the absence of either PIN2-mediated auxin transport or *TIR1/AFB*-dependent auxin perception seems insufficient for efficient establishment of gravitropic root growth.

**Brassinolide modulates PIN2 sorting upon gravistimulation.** Gravitropic root bending coincides with a transient PIN2 gradient, with more PM-resident PIN2 found at the lower side of bending roots[12]. Before completion of root bending this gradient

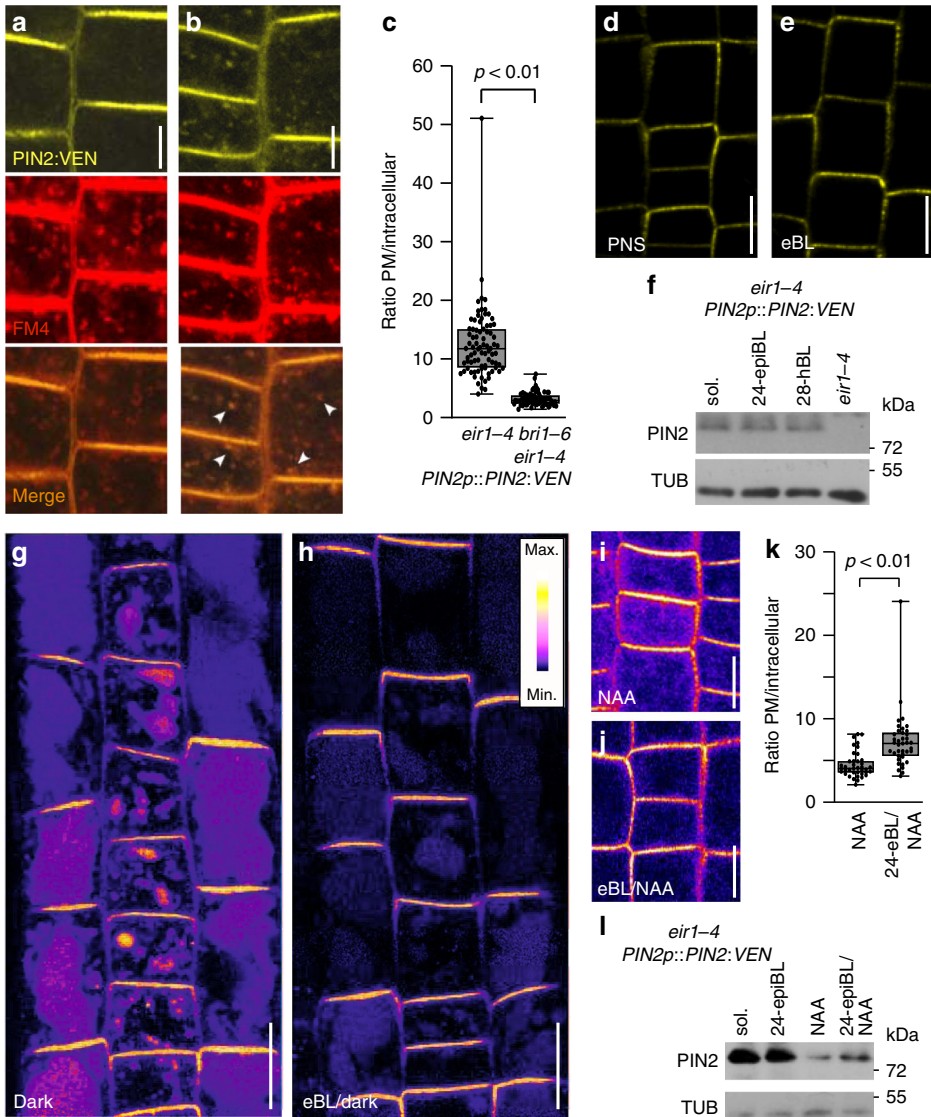

**Fig. 4 Sorting of tagged wild type PIN2 is modulated by brassinolide. a, b** Top panels: PIN2:VEN distribution in 6 DAG day-old *eir1-4 PIN2p::PIN2:VEN* (**a**) and *bri1-6 eir1-4 PIN2p::PIN2:VEN* (**b**) root meristem epidermis cells. Middle panels FM4-64 distribution after 20 min of treatment (FM4). Bottom panels: Merged images. White arrowheads: co-localization in endocytic vesicles. **c** PIN2:VEN signal ratio between PM and endosomal vesicles in *eir1-4 PIN2p::PIN2: VEN* and *bri1-6 eir1-4 PIN2p::PIN2:VEN* root meristem epidermis cells (*n* = 8 roots and 82/72 cells/dataset). Two-tailed *t*-test analysis of resulting values revealed a significant difference. **d, e** PIN2:VEN in 6 DAG *eir1-4 PIN2p::PIN2:VEN* root epidermis cells in mock-treated samples PNS (**d**) or with 100 nM eBL for 16 h (**e**). **f** Western blots performed with *eir1-4 PIN2p::PIN2:VEN* root membrane protein fraction grown on PNS or treated with 100 nM eBL or 28-homoBL for 16 h. Membrane protein extract from *eir1-4* served as control. **g, h** Heat map of *eir1-4 PIN2p::PIN2:mCherry* root meristem epidermis cells at 6 DAG, seedlings were grown with roots in the dark, either on control medium (**g**) or treated with 100 nM eBL for 3 h (**h**). **i, j** Heat map of PIN2:VEN distribution in epidermis cells of 6 DAG *eir1-4 PIN2p::PIN2:VEN*, treated with 10 µM NAA for five hours (**i**) or pre-treated with 100 nM for 16 h before NAA treatment (**j**). **k** Box plot, displaying PIN2:VEN signal distribution after NAA and eBL/NAA treatments (see '**i**, **j**'; *n* = 5 roots and 39 cells/dataset). **l** Western blots performed with *eir1-4 PIN2p::PIN2:VEN* root membrane protein fraction treated with 10 µM NAA for 5 h (NAA) or pre-treated with 100 nM eBL for 16 h before NAA treatment (24-epiBL/NAA). Material from mock-treated (sol.) and eBL-treated seedlings (24-epiBL), served as controls. Anti-α-tubulin (TUB) was used as loading control. Whiskers in box plots represent the entire range of outliers; gray boxes: first and third quartiles; center line: median; dots: values obtained. Two-tailed *t*-test analysis has been performed to test for p-values. Scale bars: a, b = 5 µm; d, e, g-j = 10 µm. Source data are provided as Source Data file.

vanishes, signifying that dynamic variations in PIN2 sorting shape differential auxin flow from the root tip into the root elongation zone[12,46]. To determine, how brassinolide might influence PIN2 distribution in gravistimulated roots, we tested PIN2:VEN distribution over time. In *eir1-4 PIN2p::PIN2:VEN* controls, a lateral PIN2 gradient with more Venus signals at the lower side of the meristem became visible after 90' and 150', and

disappeared after extended gravistimulation (Fig. 6a–e). Roots pre-treated with brassinolide did not establish a comparable PIN2 gradient, indicating that brassinolide interferes with differential PIN2 distribution in gravistimulated roots (Fig. 6a–e).

The apparent eBL effects on PIN2:VEN upon gravistimulation, hint at participation of brassinolide signaling in sorting control of PIN2 in gravity-responding roots. To visualize endogenous

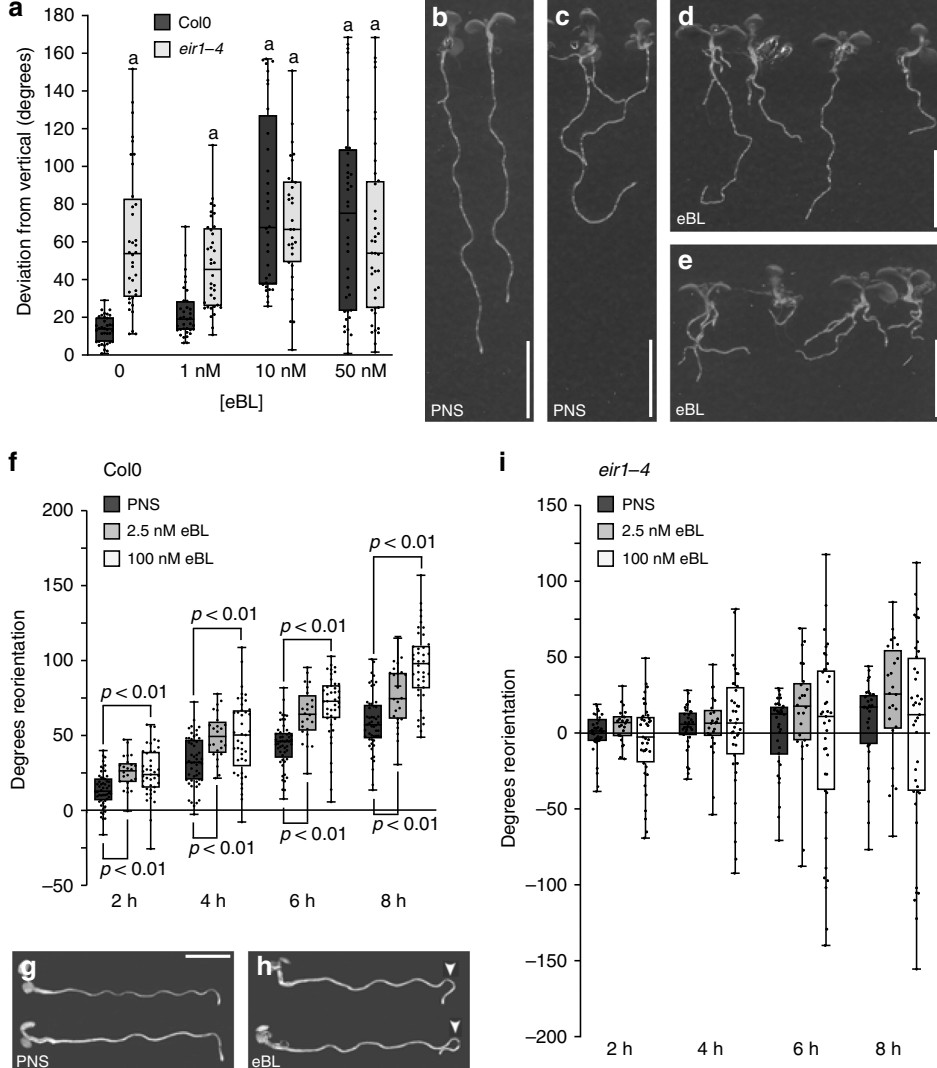

**Fig. 5 Crosstalk between brassinolide and PIN$_2$ in directional root growth. a** Deviation of root growth from vertical of 6 DAG Col0 and *eir1-4* germinated in the presence of indicated eBL concentrations. 30-38 roots were analyzed for each dataset. **b–e** Vertically oriented 6 DAG Col0 (**b, d**) and *eir1-4* (**c, e**) seedlings germinated on PNS (**b, c**) or 10 nM eBL (**d, e**). **f** Root reorientation of 6 DAG Col0 primary roots incubated on the indicated growth medium. 24-51 roots were analyzed for each dataset. **g, h** 6 DAG Col0 seedlings, after o/n gravistimulation on PNS (**g**) and on 100 nM eBL (**h**). White arrowheads indicate root overbending. **i** Root reorientation of 6 DAG *eir1-4* primary roots incubated on the indicated growth medium. 24-47 roots were analyzed for each dataset. One-way ANOVA with post-hoc Tukey HSD test was used to determine *p*-values. 'a' indicates significant differences ($p < 0.01$) to Col0 control roots. Whiskers in box plots cover the entire range of outliers obtained in the datasets; gray boxes display first and third quartiles; center line: median; dots: values obtained. Scale bars: b–e, g, h = 10 mm. Source data are provided as Source Data file.

brassinolide signaling upon gravistimulation, we made use of brassinolide-responsive *35S::BES1:GFP*[47]. Specifically, reporter stabilization and increased nuclear localization in response to brassinolide, makes this line a versatile tool for studying variations in brassinolide signaling[47] (Supplementary Fig. 10a–c). Quantification of BES1:GFP in nuclei of root meristem epidermis cells revealed establishment of a transient expression gradient in horizontally positioned roots. After 90 min of gravistimulation, we observed a transient increase in BES1:GFP signal intensities at the root's lower side, when compared to the upper side (Fig. 7a–d). In contrast, no BES1:GFP gradient was established on eBL medium, indicating that exogenously applied eBL interferes with differential brassinolide signaling in gravistimulated roots (Supplementary Fig. 10d–f).

Taken together, our experiments revealed asymmetric brassinolide signaling in gravistimulated roots, coinciding with a lateral PIN2 gradient. Furthermore, disruption of the lateral PIN2

gradient by brassinolide treatment is in line with a role for endogenous brassinolide signaling in the spatiotemporal control of PIN2 in gravistimulated roots.

**Modeling auxin distribution in dependence of PIN2.** Current models suggest that elevated PIN2 levels at the lower side of gravity-responding roots cause differential auxin flow into the root elongation zone to stimulate altered root elongation[12,46]. We found that brassinolide treatment, whilst interfering with PIN2 gradient formation in gravistimulated roots, does not block differential cell elongation and root bending. We therefore asked, if brassinolide affects differential auxin distribution/signaling, which functions as a prerequisite for root gravitropism, and tested intensity of auxin-responsive *R2D2*[40]. In mock-treated controls, a *D2* signal gradient was clearly visible after 90 min of gravistimulation, which disappeared at later time points, reflecting

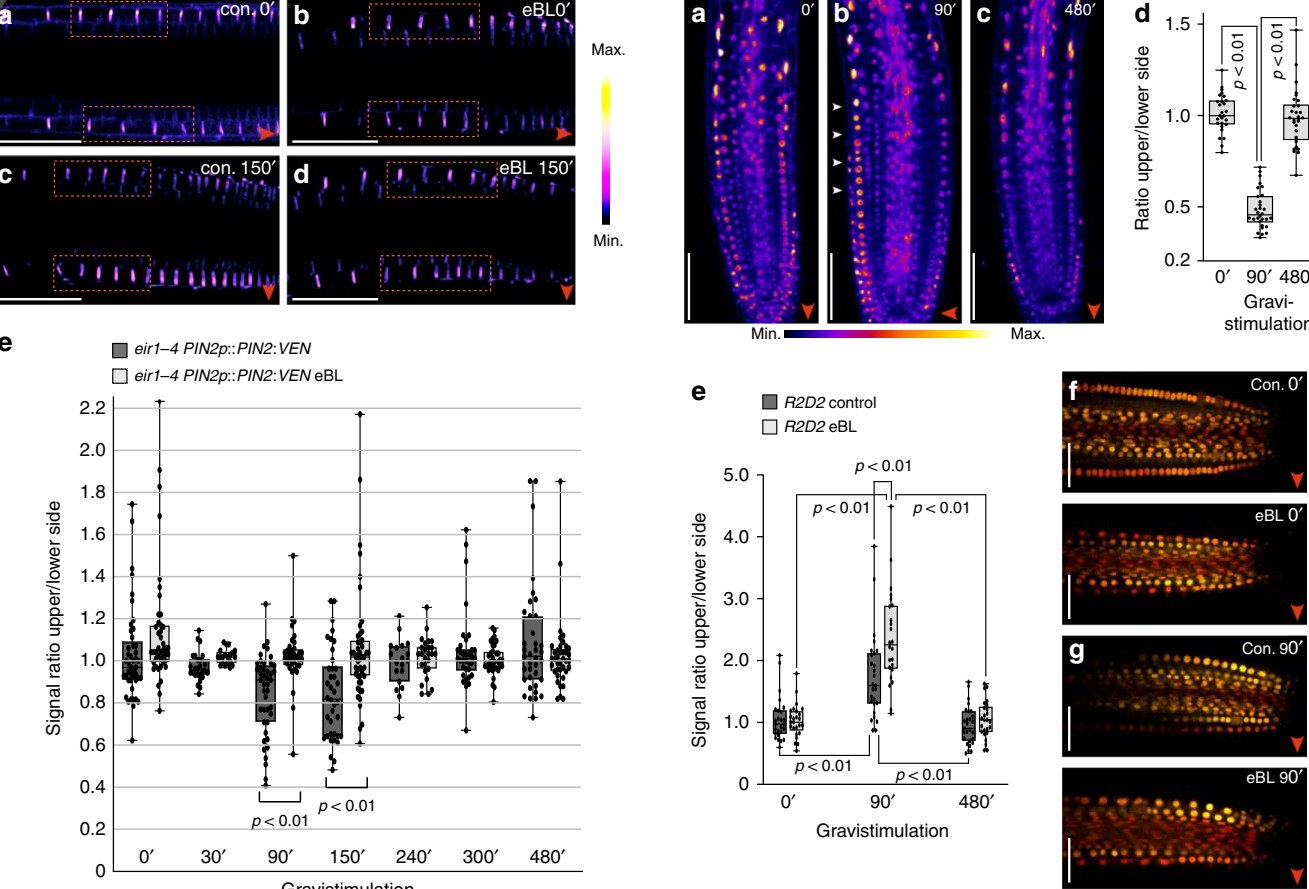

**Fig. 6 PIN2 expression in gravistimulated roots is controlled by brassinolide. a–d** Stacks of z-sections generated for analysis of PIN2:VEN signal distribution (see 'e'), at time point zero (**a**, **b**) and after 150 min of gravistimulation (**c**, **d**). Relative signal intensities were determined measuring signals in adjacent root epidermis cells in the bending zone (highlighted by dotted red rectangles). Red arrowheads: direction of gravity vector. **e** PIN2:VEN signal ratios in epidermis cells at upper vs. lower side of gravistimulated root meristems. 19–47 roots were analyzed for each dataset. One-way ANOVA with post-hoc Tukey HSD test revealed significant differences between control roots on PNS and roots pre-treated with 100 nM eBL, after 90 and 150 min of gravistimulation. Whiskers in box plots cover the entire range of outliers obtained in the datasets; gray boxes display first and third quartiles; center line: median; dots: values obtained. Scale bars: a, b = 50 μm. Source data are provided as Source Data file.

**Fig. 7 Brassinolide and auxin responses in gravistimulated roots. a–c** Heat map of *35S::BES1:GFP* expression in root meristems after 0 (**a**), 90 (**b**) and 480 (**c**) min of gravistimulation. White arrowheads indicate asymmetric reporter expression after 90 min. **d** BES1:GFP signal ratios at the upper vs. lower side of gravistimulated roots. Signal intensities in 9–10 roots were determined for each dataset. **e** R2D2 signal ratios at the upper vs. lower side of gravistimulated root meristems, grown on PNS or pre-treated with 100 nM eBL. 24-27 roots of 6-day-old seedlings were analyzed for each dataset. **f, g** R2D2 expression and gradient formation in control roots and eBL-treated roots, after 0 (**f**) and 90 (**g**) minutes of gravistimulation. One-way ANOVA with post-hoc Tukey HSD tests were performed, to obtain *p*-values. Whiskers in box plots cover the entire range of outliers obtained in the datasets; gray boxes: first and third quartiles; center line: median; dots: values obtained. Scale bars: a–c, f, g = 50 μm. Red arrowheads: Direction of gravity vector. Source data are provided as Source Data file.

differential auxin signaling in gravity-responding roots during a defined stage of root curvature (Fig. 7e–g)[48]. Gravistimulated *R2D2* roots pre-treated with brassinolide also established a discernible signal gradient, even slightly more pronounced than in controls (Fig. 7e–g). From that it seems that, unlike suggested in current models, formation of a PIN2 gradient is not categorically required for differential auxin signaling in gravistimulated roots.

To clarify this conundrum, we took a simulation approach, and analyzed auxin distribution in dependence of variable PIN2 levels, by extending a recently developed model of auxin transport (Supplementary Note 1, Supplementary Fig. 11a)[49]. Intracellular auxin concentrations are governed by three distinct transport mechanisms across cellular membrane boundaries, namely diffusion as well as active transport by influx and efflux carriers[12,50]. Cumulative permeabilities for auxin at each membrane boundary are jointly defined by these three parameters

and thus, should be influenced by variations in any of these determinants[48]. To model a PIN2 gradient and effects on intracellular auxin concentration, we varied auxin permeability in proportion to relative PIN2 abundance, as determined experimentally (Fig. 6e).

PIN2-mediated auxin permeability on the opposing sides of the root is denoted by $P_{\mathrm{PIN2}}^{\mathrm{lower\,side}}$ and $P_{\mathrm{PIN2}}^{\mathrm{upper\,side}}$ (Fig. 8a, b schemes, blue bars, Supplementary Fig. 11a). In the symmetrical (initial) state, there is no difference in permeabilities on both sides: $P_{\mathrm{PIN2}}^{\mathrm{lower\,side}} = P_{\mathrm{PIN2}}^{\mathrm{upper\,side}} = P_{\mathrm{PIN2}}^{\mathrm{initial}}$. The ratio of PIN2 permeabilities is assumed proportional to the measured PIN2:VEN signal ratio, as presented in Fig. 6e: $\frac{P_{\mathrm{PIN2}}^{\mathrm{upper\,side}}}{P_{\mathrm{PIN2}}^{\mathrm{lower\,side}}} = \frac{\mathrm{PIN2:Venus\,signal\,upper\,side}}{\mathrm{PIN2:Venus\,signal\,lower\,side}} = k$. We used this ratio, denoted as parameter $k$, to define permeabilities as

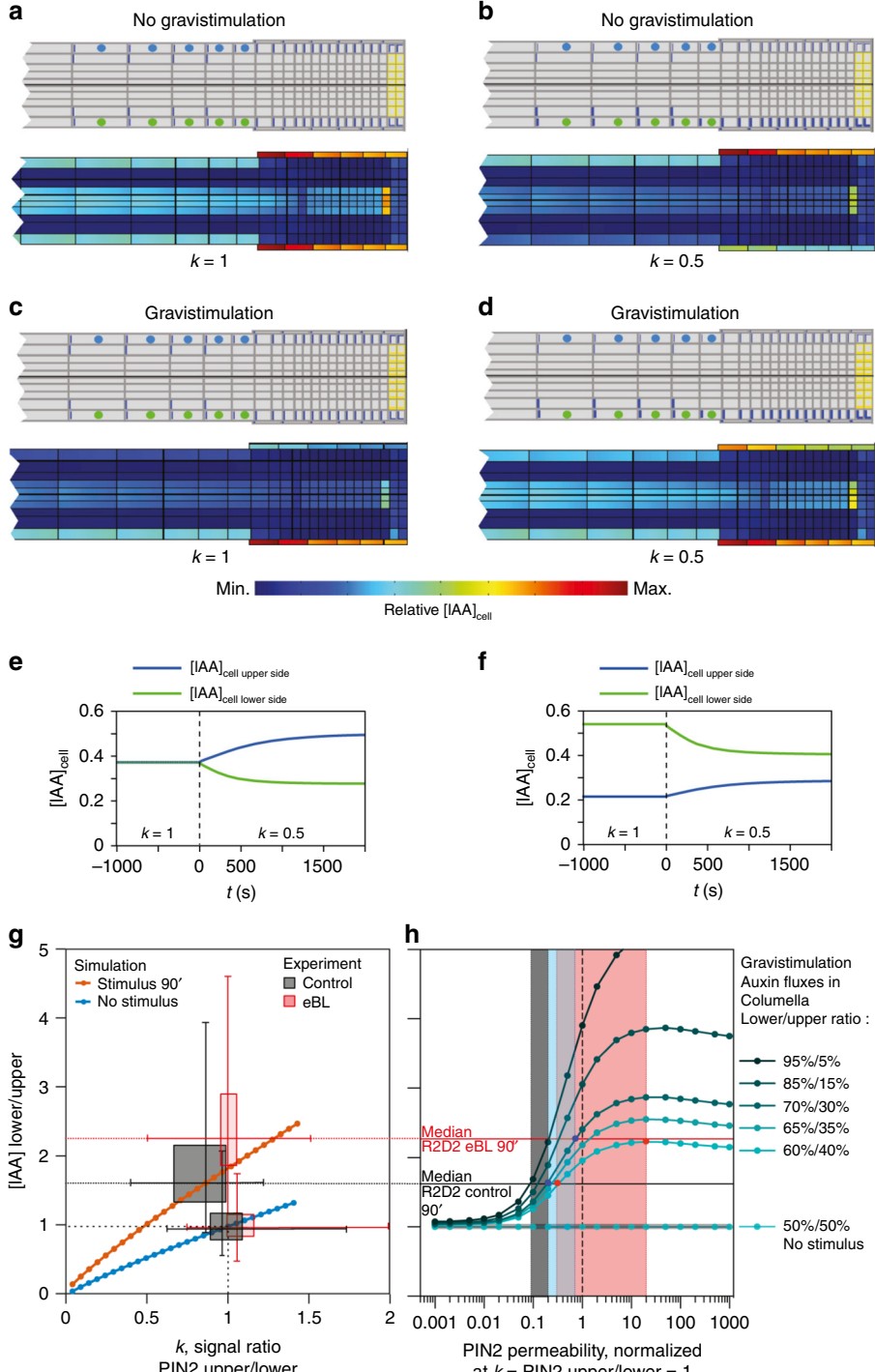

$$P_{PIN2}^{upper\,side} = P_{PIN2}^{initial} \cdot \sqrt{k}, \tag{1}$$

$$P_{PIN2}^{lower\,side} = \frac{P_{PIN2}^{initial}}{\sqrt{k}}. \tag{2}$$

In our experiments, gravistimulated control roots exhibited PIN2 asymmetry with a minimum median value $k = 0.81$, whereas eBL-treated samples exhibited no asymmetry ($k$ close to 1) (Fig. 6e). We calculated $[IAA]_{cell}$ distribution for a range of

$k$ values (from 0.05 to 1.4) and determined how PIN2 asymmetry affects $[IAA]_{cell}$, and how eBL treatment could influence $[IAA]_{cell}$.

First, we tested conditions under which initial $[IAA]_{cell}$ is equal at upper and lower side of the root elongation zone, and no asymmetric $[IAA]_{cell}$ is present in root tip cells (Fig. 8a, b, e). Upon building asymmetry in PIN2 permeability we observed an increase in $[IAA]_{cell}$ at the side with lower PIN2 abundance (Fig. 8b, e): with an asymmetry coefficient $k = 0.81$, we obtained $\frac{IAA^{lower\,side}}{IAA^{upper\,side}} \cong 0.8$ (calculated for epidermal cells of the elongation zone; Fig. 8b, g; blue curve). A linear relationship between $\frac{P_{PIN2}^{upper\,side}}{P_{PIN2}^{lower\,side}}$

**Fig. 8 Simulation of auxin distribution in root tips. a–d** Top: Model geometry showing PIN2-mediated permeabilities (blue lines) and PIN3/PIN7 permeability (yellow lines). Thickness of the lines represents permeability values. Bottom: Corresponding steady-state intracellular auxin concentrations $[\text{IAA}]_{\text{cell}}$. **a** Symmetric PIN2 and PIN3/PIN7 permeabilities distribution. **b** PIN2 permeability is reduced at the upper side and increased at the lower side, while PIN3/PIN7 permeabilities are symmetric (no gravistimulation). **c** Symmetric PIN2 distribution in the case of asymmetric PIN3/PIN7 permeabilities (gravistimulation). **d** PIN2 permeability is reduced at the upper side and increased at the lower side, and PIN3/PIN7 permeabilities are also asymmetric (gravistimulation). **e, f** Evolution of $[\text{IAA}]_{\text{cell}}$ at the upper (blue) and lower (green) side in the EZ, starting with equal PIN2 distribution ($k = 1$), and after PIN2 asymmetry is established at time point zero ($k = 0.5$). Plotted is an average $[\text{IAA}]_{\text{cell}}$ of the first 5 cells of the EZ (highlighted by blue and green dots in 'a'). No gravistimulation: Initial $\frac{\text{IAA(lower side)}}{\text{IAA(upper side)}} = 1$ (**e**). Gravistimulation: Initial $\frac{\text{IAA(lower side)}}{\text{IAA(upper side)}} = 2.3$ (**f**). **g** Simulation of $\frac{\text{IAA(lower side)}}{\text{IAA(upper side)}}$ in dependence of PIN2 asymmetry $k$ in the case of no gravistimulation (blue line; $\frac{\text{IAA(lower side)}}{\text{IAA(upper side)}} = 1$ at $k = 1$), and after gravistimulation under control conditions (orange line; $\frac{\text{IAA(lower side)}}{\text{IAA(upper side)}} = 1.62$ at $k = 0.81$). Boxes display experimental values determined for $\frac{\text{IAA(lower side)}}{\text{IAA(upper side)}}$ and $k$ for control and eBL-treatement (see text and Figs. 6c and 7e). Whiskers represent the range of values obtained in the datasets. **h** Dependence of $\frac{\text{IAA(lower side)}}{\text{IAA(upper side)}}$ on initial PIN2 permeability in gravistimulated roots, for $k = 1$ and variable gravistimulation strengths (normalized by default value: $P_{\text{PIN2}}^{0\,\text{initial}} = 0.5\,\text{um/s}$). Gravistimulation strength is defined as the ratio of auxin flux towards the lower side vs. the upper side in columella cells (in %). Color shadings highlight shifts in PIN2 permeability required to increase $\frac{\text{IAA(lower side)}}{\text{IAA(upper side)}}$ from measured control values (1.62, black line) to eBL-treatment values (2.3, red line) under: high gravistimulation strength (auxin flux ratio 95%/5%, gray), intermediate (auxin flux ratio of 70%/30%, blue) and lowest possible gravistimulation strength (auxin flux ratio 60%/40%, pink). Source data are provided as Source Data file.

and $\frac{\text{IAA}^{\text{lower side}}}{\text{IAA}^{\text{upper side}}}$ is maintained, independently of initial PIN2 permeabilities, which influence only the magnitude of this positive coefficient of proportionality (Supplementary Note 14; Supplementary Fig. 16a–c). This relationship also applies, when changing permeability values of the auxin importer AUX1, a key determinant of shootward auxin transport in root meristems[50] (Supplementary Note 16, Supplementary Fig. 17b).

Next, we introduced initial $[\text{IAA}]_{\text{cell}}$ asymmetry in the root tip, reflecting auxin gradient formation mediated by asymmetric accumulation of PIN3 and PIN7 at the lower side of columella cells[51] (Fig. 8c, d, f). We simulated this gravistimulation case by increasing permeability at the lower side of columella cells, so that $\frac{\text{IAA}^{\text{lower side}}}{\text{IAA}^{\text{upper side}}} = 1.62$ at $k = 0.81$. This value was chosen to match experimental values at 90′ after gravistimulation in control roots ($\frac{\text{IAA}^{\text{lower side}}}{\text{IAA}^{\text{upper side}}} = \frac{\text{R2D2(upper side)}}{\text{R2D2(lower side)}} = 1.62$, $k = 0.81$ Fig. 6e; Fig. 7e). Keeping permeability distribution in columella cells unchanged, we calculated $\frac{\text{IAA}^{\text{lower side}}}{\text{IAA}^{\text{upper side}}}$ depending on $k$ (plotted on Fig. 8g, orange line), revealing that PIN2 asymmetry with higher permeability at the lower side effectively decreases the $[\text{IAA}]_{\text{cell}}$ ratio (Fig. 8d, f, g). In related simulations, we tested modifications in overall root morphology, which occur in response to brassinolide treatment[23,28] (Supplementary Note 10; Supplementary Fig. 12), and assessed consequences of altered PIN2 localization (Supplementary Note 11; Supplementary Fig. 14). These variations did not strikingly modify the values of $[\text{IAA}]_{\text{cell}}$ ratios, obtained in our simulations (Supplementary Fig. 12c; Supplementary Fig. 14e).

We asked, if a loss of PIN2 asymmetry upon eBL treatment is solely responsible for increased $\frac{\text{IAA}^{\text{lower side}}}{\text{IAA}^{\text{upper side}}} = 2.28$ (Fig. 7e). When mimicking no PIN2 asymmetry ($k = 1$) upon gravistimulation, we obtained $\frac{\text{IAA}^{\text{lower side}}}{\text{IAA}^{\text{upper side}}} = 1.93$, which is lower than the experimental value ($= 2.28$) observed in response to eBL treatment (Fig. 8g, orange line). This discrepancy argues for additional determinants involved in auxin gradient formation under eBL treatment, and we therefore extended simulations. We assessed consequences of a range of physiologically plausible values for PIN2 and AUX1 permeabilities as well as PIN3/7 asymmetry, to estimate how such changes could influence $\frac{\text{IAA}^{\text{lower side}}}{\text{IAA}^{\text{upper side}}}$. PIN3/7 asymmetry in columella cells turned out to define gravistimulation strength, i.e. the proportion of auxin that is directed to the root's lower side during the gravitropic response. There is a linear relationship between PIN3/7 asymmetry and $\frac{\text{IAA}^{\text{lower side}}}{\text{IAA}^{\text{upper side}}}$.

Conversely, the functions of PIN2 and AUX1 permeability are not monotonical (Fig. 8h; Supplementary Fig. 15a, b and 17a–c). Specifically, we identified intermediate permeability values of PIN2 and AUX1, at which $\frac{\text{IAA}^{\text{lower side}}}{\text{IAA}^{\text{upper side}}}$ is highest (Supplementary Note 16). Upon simulation of very low PIN2 permeabilities, no auxin asymmetry is established, even under very strong gravistimulation strength, reproducing *pin2* mutant phenotypes (Fig. 8h).

Because of the nonlinear relationship between AUX1/PIN2 permeabilities and $\frac{\text{IAA}^{\text{lower side}}}{\text{IAA}^{\text{upper side}}}$, consequences of changes in PIN2 and AUX1 levels strongly depend on initial AUX1/PIN2 permeability values and on the gravistimulation strength. An at least twofold increase in overall PIN2 permeability at upper and lower side of the root is required, to account for a shift of $\frac{\text{IAA}^{\text{lower side}}}{\text{IAA}^{\text{upper side}}}$ from 1.62 (control) to 2.3 (eBL-treated). This however, is only true, upon simulating a very strong asymmetry in PIN3/PIN7 permeabilities, together with very low levels of initial PIN2 permeability (Fig. 8h, shaded gray region). If any of these minimal requirements is not fulfilled, an up to 100-fold increase of PIN2 permeability would be needed to cause such a shift in $\frac{\text{IAA}^{\text{lower side}}}{\text{IAA}^{\text{upper side}}}$ (Fig. 8h, shaded blue and pink regions). These conditions appear physiologically irrelevant, supported by our experimental data, with increases in PIN2:VEN signal intensities never exceeding a factor of 2 (Supplementary Fig. 15c). In contrast, by introducing PIN2 asymmetry into the model, an increase of PIN2 permeability values within the range of experimentally observed variations in PIN2:VEN abundance, is sufficient for the calculated shifts in $\frac{\text{IAA}^{\text{lower side}}}{\text{IAA}^{\text{upper side}}}$ (Supplementary Figs. 15a, b and 17, Supplementary Notes 13 and 16). Thus, whilst increased overall PIN2 (or AUX1) permeability influences auxin distribution, it appears insufficient for eBL-induced shifts in $\frac{\text{IAA}^{\text{lower side}}}{\text{IAA}^{\text{upper side}}}$ ratios, as it would require drastic permeability changes. In contrast, for values of PIN2 permeability similar to those reported in the literature ($0.1 \leq P_{\text{PIN2}}^{\text{initial}} \leq 10$; Supplementary Table 1), even a moderate PIN2 asymmetry ($k = 0.81$) will have substantial effects on $\frac{\text{IAA}^{\text{lower side}}}{\text{IAA}^{\text{upper side}}}$, and therefore is not negligible for our model (Supplementary Fig. 17d, compare bars # 1 & 6 or 5 & 9).

Our simulation revealed an inverse proportionality of $[\text{IAA}]_{\text{cell}}$ and PIN2 asymmetry levels in gravistimulated roots, with $\frac{\text{IAA}^{\text{lower side}}}{\text{IAA}^{\text{upper side}}} \sim \frac{P_{\text{PIN2}}^{\text{upper side}}}{P_{\text{PIN2}}^{\text{lower side}}}$. The coefficient of proportionality depends on the initial asymmetry of $\frac{\text{IAA}^{\text{lower side}}}{\text{IAA}^{\text{upper side}}}$, induced in columella cells and rises with increasing $\frac{\text{IAA}^{\text{lower side}}}{\text{IAA}^{\text{upper side}}}$ values (Fig. 8g). This can be explained by

increased PIN2 efflux activity resulting in lower cellular IAA levels, as long as other transport activities remain unchanged (Supplementary Notes 10 and 17; Supplementary Fig. 13). Reduced PIN2 activity in contrast, moves the flux balance in favor of influx, giving rise to increased intracellular auxin concentrations. According to this simulation, the effect of asymmetric PIN2 abundance on auxin distribution would be opposite to that of gravity-induced PIN3/PIN7 relocation in columella root cap cells. Consequently, it would counteract differential elongation of cell files at upper and lower sides of the root meristem, by dampening the steep auxin concentration gradient, initially established in gravistimulated columella root cap cells.

## Discussion

Variations in subcellular PIN distribution determine directional auxin transport. Hence, deciphering the molecular basis of PIN sorting is key to our understanding of mechanisms, by which these proteins influence plant growth. Here, by employing a constitutively endocytosed version of *Arabidopsis* PIN2, we present a detailed characterization of *PIN2* crosstalk with brassinolide signaling, and its functional implications for root gravitropism.

We utilized a version of PIN2, with a ubiquitin-tag serving as a signal for protein internalization and degradation, which makes this reporter highly suitable for investigating stability and sorting of endocytosed PIN2[7]. Stabilization of PIN2:ubq:VEN by brassinolide represents an example for the feasibility of this approach. This response depends on canonical brassinolide signaling via *BRI1* and on downstream GSK3/Shaggy-type protein kinases[21,37]. The effect is dosage-dependent and independent of de novo protein biosynthesis, suggestive of non-genomic mechanisms, as implicated in PIN sorting control by auxin and GA[11,14]. Alternatively, brassinolide might act indirectly, via affecting transcription of regulators of PIN sorting, as suggested for microtubule associated protein *CYTOPLASMIC LINKER-ASSOCIATED PROTEIN1 (CLASP1)*. CLASP1 promotes PIN2 recycling to the PM, presumably via adjusting the association of sorting endosomes and microtubules in a *SORTING NEXIN1-(SNX1)*-dependent manner[10]. Brassinolide in turn, causes transcriptional down-regulation of *CLASP1*, potentially modulating PIN2 recycling to the PM[30].

Next to crosstalk between brassinolide and *CLASP1*, implicated in protein recycling, our analysis highlights brassinolide-mediated inhibition of PIN2 endocytic sorting from the PM. Effects are strong in the case of constitutively endocytosed PIN2:ubq:VEN, whereas comparatively mild responses are observed with wild type PIN2 reporters. PIN2 labeled for degradation, thus likely represents a preferred target for this type of hormonal regulation. This is supported by our observations, demonstrating that brassinolide stabilizes PIN2 under conditions, promoting its vacuolar degradation. Furthermore, we found that inhibition of brassinolide biosynthesis no longer causes enhanced degradation of ubiquitylated PIN2 upon interfering with the TOL/ESCRT machinery[8], which links brassinolide signals to endocytosis of ubiquitylated PM cargo. Brassinolide-controlled variations in the distribution of PIN2 thus, seem to occur via adjustments, both in protein recycling and endocytosis, affecting PAT in the regulation of root growth in general, and gravitropism, in particular.

Coordination of root growth in response to gravity involves differential auxin flow from the root tip into the elongation zone. This asymmetry seemingly depends on redistribution of PIN proteins to the bottom side of columella root cap cells, redirecting auxin flow to the root's lower margin[51]. Enhanced auxin transport at the root's lower side, when compared to the upper side, was suggested to be consolidated further by a transient PIN2 gradient, with more PM-resident PIN2 at the lower side, resulting

from differential sorting and turnover of the protein[7,12,46]. Consequently, asymmetric auxin flow would induce differential cell elongation and gravitropic root bending. Along these lines, resetting of differential auxin transport to default levels would be promoted by getting rid of the PIN2 gradient upon completion of root bending[12,46]. Our findings demonstrate crosstalk between auxin and brassinolide signals, via PIN2 sorting in gravity-responding roots. By analogy to modes of action suggested for other growth regulators, spatiotemporal variations in brassinolide signaling influence PIN2 PM association, thereby adjusting auxin flow[12,14]. Surprisingly though, interference with PIN2 gradient formation upon eBL treatment, neither prevented gravitropic root bending, nor differential auxin signaling. Abolished PIN2 gradient formation might be compensated by—for example—local adjustments in PIN2 activity. Nevertheless, our observations demonstrate that formation of a lateral PIN2 expression gradient is not categorically required for gravitropic root bending.

Our modeling approach offers a plausible explanation for our experimental observations. We propose that the steep auxin concentration gradient, established in columella root cap cells[51], is sufficient for differential auxin flow into the root elongation zone, as long as PIN2 is functionally expressed. Our simulation suggests that formation of a lateral PIN2 gradient in gravi-stimulated roots results in a less pronounced auxin gradient, due to transiently reduced PIN2 levels at the root's upper side. Diminished auxin efflux, caused by such reduced PIN2 levels, will give rise to a noticeable reduction in the $\frac{\text{IAA(upper side)}}{\text{IAA(lower side)}}$ ratio. In contrast, equal PIN2 abundance at both sides of gravistimulated root meristems is predicted to generate a steeper auxin gradient. This is caused by efficient auxin efflux at the root's upper side, resulting in a localized reduction in cellular auxin steady-state levels. Likely consequences of steeper auxin gradients involve accentuation of differential cell elongation, which might contribute to root overbending phenotypes similar to those observed in response to brassinolide.

Timing and kinetics of differential auxin signaling and PIN2 sorting are consistent with our model. Specifically, whilst differential auxin signaling in the root meristem is manifested already after 30 min of gravistimulation[48], a PIN2 gradient becomes apparent during later phases of root bending[12,46]. At this stage, differential PIN2 abundance could already antagonize effects of asymmetric auxin flow, preventing disproportionate root bending. Such activities would be no longer required upon completion of root bending, reflected in recovery of PIN2 levels[12,46]. Furthermore, since local [IAA]$_{\text{cell}}$ adjustments in dependence of variable PIN2 expression appear to occur already within a few minutes (Fig. 8e, f), variations in abundance and localization of PIN2 could facilitate swift adjustments in auxin flow, by limiting effects of the auxin gradient established in the root tip.

Variable *BES1:GFP* expression in gravistimulated roots hints at an involvement of brassinolide signaling in the regulation of PIN2. Along these lines, deficiencies in root gravitropism that have been described for brassinolide signaling mutants[42], could reflect defects in intracellular sorting of PIN2. Together with additional root growth responses triggered by brassinolide, the plant hormone thus appears to shape root growth via adjustments in auxin flow. Integration of these distinct pathways into a network of regulatory events that jointly define adaptive growth responses, remains a challenge for future research.

## Methods

**Plant lines and growth conditions.** Plants were grown on PNS plant nutrient agar plates[52], supplemented with 1% (w/v) agar and 1% (w/v) sucrose, in a 16 h light/8 h dark regime at 22 °C. *PIN2p::PIN2:ubq:VEN*[7], *PIN2p::PIN2:VEN*[7], *R2D2*[40], *35S:: BES1:GFP*[47], and *pINTAM»RFP:HUB*[6] transgenic lines have been described. We made use of reduced pH sensitivity of the mCherry tag, and generated mCherry-

tagged PIN2 for assessment of vacuolar sorting of PIN2. *PIN2p::PIN2:mCherry* was obtained by replacing Venus in *PIN2p::PIN2:VEN* with the mCherry tag. For generation of *RP40::PIN2:VEN* and *RP40p::PIN2:ubq:VEN*, an *RP40* promoter fragment has been put upstream of the respective ORFs[32]. Constructs were transformed into *eir1-4*, using the floral dip method[53]. Resulting transgenics were selected for single copy insertions and propagated to homozygosity. *Eir1-4, bri1-6, tir1-1 afb1-1 afb2-1 afb3-1*, and *det2-1* mutant lines have been described elsewhere[12,33,45,54]. For expression analysis, reporter lines were introduced into the respective plant lines by crossing. Resulting progeny was then grown into F3 to homozygosity. Isogenic mutant and wild type lines resulting from these crosses were used for comparison of reporter expression. For testing the effect of eBL on PIN2:mCherry distribution in dark grown roots, seedlings were grown accordingly[55]. Ubiquitylation-deficient *eir1-4 PIN2p::pin2^{K12R}:VEN* and *eir1-4 PIN2p:: pin2^{K17R}:VEN* lines were considered for growth assays as well, as stabilization of PIN2 resulting from diminished ubiquitylation, resembles eBL effects on wild type PIN2[7]. In planta analysis of these lines, however, indicated that functionality of the *pin2^{K-R}* alleles differs from that of wild type *PIN2*, which made a comparison of growth and hormone responses unfeasible (Konstantinova and Luschnig, in preparation).

**Chemicals, pharmacological, and growth assays**. All chemicals used in this study were of analytical grade and were stored as 1000 × stock solutions (24-epibrassinolide, 28-orthobrassinolide, stigmasterol, 1-NAA, PAC, 4-hydroxytamoxifen were dissolved in ethanol, whereas DMSO was used for BFA, WM, BRZ, bikinin, cholesterol, DEX). Upon assaying, compounds were added to liquid ½ × Murashige Skoog growth medium to give the indicated working concentrations. Controls were treated with the corresponding amounts of solvent only. For quantitative analysis on Western blots, plant material was grown on solid PNS medium for the indicated period of time. For short term treatment with plant growth regulators and inhibitors, seedlings grown on the surface of solid medium were sprayed with the compounds diluted in liquid growth medium. After incubation for the time indicated, root material was harvested, snap frozen in liquid nitrogen followed by protein extraction. For induction of *pINTAM»RFP:HUB* plant material was germinated on PNS, followed by induction in the presence of 2 μM 4-hydroxytamoxifen for 24 h. Induced material, together with non-induced controls was analyzed at 5–6 DAG.

For root gravitropism assays, seedlings were germinated on regular growth medium for the indicated period of time. Seedlings were then transferred onto fresh control plates or onto plates containing indicated concentrations of eBL. After o/n equilibration in the dark, seedlings were aligned in a horizontal orientation. Plates were then turned clockwise at an angle of 90° and kept in darkness to minimize light-mediated effects on root growth. Plates were scanned at indicated time points and images were used for determination of root reorientation, employing the angle tool of ImageJ (NIH). The root tip angle of individual roots at time point zero was defined as 0°. Growth reorientation of these individual roots was then followed over time, and defined as positive in the case of growth in the direction of the gravity vector, and as negative, when growing in the opposite direction. For root growth assays, seedlings of each genotype were germinated on PNS in the presence of the indicated drugs. After incubation on vertically positioned nutrient plates, seedlings were scanned and root angles were determined as deviation from vertical, using ImageJ. For root waving assays, seedlings of each genotype were germinated on vertically oriented nutrient plates. Seedlings were scanned and resulting images used for determination of root waves per root length, using ImageJ.

**Microscopy**. CLSM images were generated using a Leica SP5 (Leica Microsystems, Wetzlar, Germany) microscope. For imaging, we used the following excitation conditions: 488 nm (GFP), 514 nm (Venus), 561 nm (mCherry, RFP and FM4-64). For endocytic sorting studies, seedlings were transferred from horizontally oriented nutrient plates into 6-well plates with liquid medium and incubated in the presence of FM4-64 (Invitrogen; working concentration 2 μM) for 30 min before CLSM visualization. For visualization of *35S::BES1:GFP* in gravistimulated seedlings, we viewed root meristems at indicated time points. Relative gray values in the nuclei of 4–5 adjacent epidermis cells in the cell division zone were determined at the upper/lower side of incubated seedlings ImageJ/Fiji software. The average of these values was then used for determination of the signal ratio. For analysis of intracellular BES1:GFP signal distribution, we determined the signal intensities in nuclei and the cytoplasm of columella root cap cells and calculated the corresponding ratio.

For assessment of PIN2 reporter signal distribution we determined the ratio of PM-localized vs. intracellular reporter signals, by determining gray values in areas of identical size and shape. In the case of BFA and WM treatments, signals were determined in intracellular compartments that were induced by drug treatment and compared to signal intensities at the PM, by using areas of identical size and shape.

For determination of PIN2 Venus signals in gravistimulated roots, we used 5–6-days-old *eir1-4 PIN2p::PIN2:VEN* seedlings. Seedlings were transferred onto fresh PNS plates or onto PNS supplemented with 100 nM eBL, straightened, using fine tweezers and positioned in a horizontal orientation. After o/n equilibration and synchronization of directional root growth, seedlings were turned clockwise at an angle of 90°, followed by incubation in the dark. Samples were taken at indicated time points after gravistimulation, and subject to CLSM analysis, with laser intensities set to a minimum, in order to avoid saturation effects, upon signal quantification. For CLSM imaging we used z-sectioning, generating stacks

consisting of five slices each, which were taken in a distance of 2 μm. Stacks were used for generation of maximum projections, using the ImageJ/Fiji software. This was followed by determination of relative gray values at the PM of adjacent epidermis cells in the root elongation zone at the upper and lower side of the root meristem (4–5 on each side). Resulting averages of signals at the upper and lower side were then used for determination of signal ratios in individual roots.

For imaging the effect of eBL on PIN2:mCherry distribution in dark grown roots a Zeiss LSM 880 confocal microscope (Carl Zeiss, Germany) equipped with C-Apochromat ×40/1.2 W objective was used. Fluorescence signals were processed with the Zen Blue software (Zeiss) where PIN2:mCherry distribution was evaluated as ratio of mean fluorescence intensity at the apical PM to mean intracellular fluorescence intensity of individual trichoblast cells. Average values were depicted as box plots, their statistical significance was calculated using two-tailed *t*-test in Sigma Plot (Systat, Chicago, IL, USA).

**RNA and protein analyses**. qRT-PCR analysis was performed using a Biorad CFX96 Real time system with the IQ SYBRgreen super mix (Biorad) according to manufacturers' recommendations. Reactions were heated for 3 min to 95 °C to activate hot start Taq DNA polymerase, followed by 40 cycles of denaturation for 10 s at 95 °C, annealing for 30 s at 55 °C and extension for 30 s at 72 °C. Expression levels were normalized to the expression levels of *EF1a* using the Livak method[56]. Primers pairs used for PIN2 reporters were 5′-ATTGCTTAGGGCGATGTACG-3′ and 5′-T AATTGAACCAGCCGTCTCC-3′. For amplification of the reference gene *EF1a* (At1g07940) 5′-TGAGCACGCTCTTCTTGCTTTCA-3′ and 5′-GGTGGTGGCATC CATCTTGTTACA-3′ were used.

For membrane protein extraction, root material was homogenized and resuspended in extraction buffer (50 mM Tris pH 6.8; 5% (v/v) glycerol; 1.5% (w/v) insoluble poly-vinylpolypyrrolidone; 150 mM KCl; 5 mM Na EDTA; 5 mM Na EGTA; 1 mM DTE; 50 mM NaF; 20 mM beta-glycero phosphate; 1 mM benzamidine; 1 mM PMSF; 3.5 μg/ml E64; 1 μg/ml pepstatin; 1 μg/ml leupeptin; 1 μg/ml aprotinin; 1 Roche Complete Mini Protease Inhibitor tablet per 10 ml)[57]. Samples were cleared by centrifugation (2100*g* for 2 min). The supernatant was saved and the pellet was re-extracted. Samples were again cleared, and combined supernatants were centrifuged (14,000*g* for 2 h) to yield total membrane pellets. For Western blot analysis, membrane pellets were dissolved in sample buffer [0.5% (w/v) CHAPS, 3% (w/v) SDS, 30% (v/v) glycerol, 60 mM dithioerythritol, 50 mM Tris (pH 6.8), 1 mM PMSF, and 0.5x Roche Complete Mini Protease Inhibitor Mixture Tablets], separated by SDS/urea PAGE, transferred to nitrocellulose membranes, and probed with affinity-purified rabbit anti-PIN2 (1:500)[12], followed by HRP-conjugated goat-anti-rabbit IgG (1:20,000; Jackson, 111-036-003). For determination of protein amounts we used mouse anti-α-TUB (clone B-5-1-2; 1:1,000; Sigma Life Sciences, T6074), followed by HRP-conjugated goat-anti-mouse IgG (1:20,000; Jackson, 115-035-003).

**Reporting summary**. Further information on research design is available in the Nature Research Reporting Summary linked to this article.

## Data availability

The authors declare that the data supporting the findings of this study are available within the paper, and its Supplementary Information files. A reporting summary for this Article is available as a Supplementary Information file. The datasets generated and analyzed during the current study are available from the corresponding author upon request. The source data underlying Figures 1e–j; 2d, g, h, o; 3d–g, i, j; 4c, f, k, l; 5a, f, i; 6e and 7d, e, as well as Supplementary Figures 1i; 2c; 3d; 4e; 5d; 6e; 7e–i, l; 9a–c; 10c, f and 15c are provided as a Source Data file. Modeling data underlying Fig. 8g, h; Supplementary Figures 12c; 13d; 14e; 15a, b; 16a, d; 17a–d are provided as a Source Data file. Model Comsol Multiphysics files with solutions are available under the following links: https://seafile.ist.ac.at/d/8d2375fedc2d4a86a471/ and https://doi.org/10.6084/m9.figshare.10279448.

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

## Acknowledgements

The authors would like to thank Brigitte Poppenberger, Elena Feraru, Jenny Russinova, Richard Strasser, Mark Estelle, Dolf Weijers and Jiri Friml for sharing published materials. This work has been supported by grants from the Austrian Science Fund (FWF P25931; P31493 to Christian Luschnig; M2379-B28 to Maria Akhmanova and Daria Siekhaus), by the Ministry of Education, Youth and Sports of Czech Republic from European Regional Development Fund 'Centre for Experimental Plant Biology': Project no. CZ.02.1.01/0.0/0.0/16_019/0000738, the Czech Science Foundation (19-13375Y), Czech Academy of Sciences (MSM200381701) and by a Docfforte fellowship from the Austrian Academy of Sciences (ÖAW) to Katarzyna Retzer. The IEB CAS Imaging facility is supported by EU OPPC CZ.2.16/3.1.00/21519 and MEYS Czech Republic CZ.02.1.01/0.0/0.0/16_013/0001775. We would like to express our gratitude to Barbara Korbei for constructive comments on the manuscript.

## Author contributions

K.R., N.K., and J.L. generated constructs and analyzed plant lines, K.R., J.L., K.M., and C.L. determined protein localization and abundance. M.A. generated the PIN2 simulations. K.R. and C.L. conceived experiments, J.P., M.A., and C.L. wrote the manuscript.

## Competing interests

The authors declare no competing interests.
