## [Peer Review File · Nature Communications]

Reviewers' comments:

Reviewer #1 (Remarks to the Author):

Previous studies, including works from the Luschnig lab, demonstrated that dynamic sorting of PIN2 between subcellular compartments, in the upper and the lower part of the bending root guides differential distribution of auxin and hence differential growth in response to gravistimulation. The level of the PIN2 protein is also modulated by brassinosteroid with no apparent effect on transcription. Few studies also demonstrated that high BR enhances root response to gravity.

Retzer et al provide new insight into the mechanism by which BR modulates PIN2 and propose an alternative model for the role of PIN2 in gravitropism. The authors took advantage of a transgenic line (*eir1-4 PIN2::PIN2:ub:Venus*) where PIN2 is constitutively ubiquitinated and thus primarily sorted for vacuolar degradation. Using this line, the authors convincingly demonstrated that BR and its canonical pathway (BRI1/BIN2) prevents vacuolar targeting and degradation of PIN2. Interestingly, it was also demonstrated that BL overrides the differential accumulation of PIN2, but the auxin gradient (as shown by the DII Venus marker) and root bending is still maintained. The authors therefore concluded that PIN2 is not necessary for gravitropic root bending. Assisted by a simulation for auxin transport, the authors propose an alternative model, where the role of differential PIN2 abundance is to counteract root bending.

Overall, this is an interesting study on one of the fundamental plant responses. However, the following aspects of this work remained somewhat unclear:

1. The authors performed several experiments, demonstrating that BL enhances PIN2 abundance in the plasma membrane by counteracting the constitutive vacuolar targeting and degradation of PIN2. Thus, exposure to BL is functionally equivalent to loss of PIN2 ubiquitylation, as in the *pin212K-R*, a construct previously made and characterized by the authors using similar experiments (PNAS, 2012). *pin212K-R*, an active transporter, failed to complement the *pin2* mutant in physiological assays (PNAS, 2012). I therefore wonder: does this mutation and high BL have a similar effect on gravitropism? (also, is *pin212K-R* levels insensitive to BRZ?).
2. Regarding the model: The auxin gradient of gravistimulated root is maintained in the presence of high BL, despite similar PIN2 sorting dynamics. Can the authors exclude the possibility that the enhanced curvature is simply due to the interaction between BL and auxin activity on target genes that further promote cell elongation in the upper side?
3. After hours of incubation with BL, cell elongation starts closer to the root tip (see also images in Figure 7h, of note: how long the roots were pre-treated with BL?). Will the simulation be affected by taking into consideration the distance from the root tip?
4. The accumulation of BES1 in the lower side is a very interesting observation. Does exposure to BL (or when in *bri1* mutant) abolish the differential accumulation BES1?
5. Figure 6h: Over-response to gravistimulus by high BL was previously described in the following missing references: Involvement of Brassinosteroids in the Gravitropic Response of Primary Root of Maize, Kim et al, Plant phys, 2000; Brassinosteroids Stimulate Plant Tropisms through Modulation of Polar Auxin Transport in Brassica and Arabidopsis, Li et al, Plant Cell, 2005.
6. Root elongation of *pin2* is shown here to be more sensitive to inhibition by BR. Could the authors discuss why?
7. Line 80: add Figure S2 d
8. Figure 1 and 2 – Indicate the number of roots used for the analysis
9. Figure 2h and I - what was the incubation time?
10. Figure 5C: add *PIN2::PIN2:Venus*
11. Western blot showed no increase in PIN2 levels in response to BL, but in Figure 7B it appears that the levels of PIN2 are enhanced by BL in the plasma membrane (compare to 0 to 150 min).

Reviewer #2 (Remarks to the Author):

1. The manuscript by Retzer et al. describes a comprehensive series of experiments aimed to link PIN2 PM stability mediated by BR signaling to affect root gravitropism. I have a number of comments on the work.
2. The construct described as PIN2:Ub:Venus is introduced as being previously reported and characterized by Leitner et al. 2012. In that publication a construct is described that is referred to as PIN2:ubq:VEN and I assume that this is the same construct used here, so it would be good to indicate that and why the name has changed since. But what is somewhat more problematic is that PIN2:ubq:VEN is described as not being able to complement eir1-4 (Fig.4I and J and description in the text of Leitner et al. 2012). Rightfully so, this was used as a strong argument to underscore the importance of ubiquitination in PIN2 endocytosis. If the constructs are the same, then it should be clearly stated that we are dealing also in the present work with a non-functional reporter in the mutant background. If not, a brief recap should introduce the new construct and state why the authors actually use the eir1-4 PIN2::PIN2:Ub:Venus background instead of wt in Fig. 2 and thereafter. This may or may not have impact on the interpretation of the experiments in this work, because to me it appears to be a somewhat artificial experimental system.
3. The next series of experiments (Fig.2-Fig.4) deal with demonstrating the involvement of the BR signaling system in the observed stabilization of PIN2:Ub:Venus at the PM. This is all well done and various tests are employed. One control that I suggest including is to determine whether the eir1-4 PIN2::PIN2:Ub:Venus lines show any change in BL insensitivity compared to wt and the eir1-4 mutant. This would confirm whether the treatments used already start at another point in the existing steady state of BR signaling.
4. In the section starting at line number 175 it is misleadingly announced that wild type PIN2 is now investigated whereas the text and figures indicate that it is rather PIN2:Venus. Here I am getting somewhat confused by the results presented in Fig.5 and S6, because when using the PIN2::PIN2:Venus construct, again assuming it is the same as reported previously, that is complementing the eir1-4 mutation there is very little effect to be seen after application of BL. So how meaningful are all of the reported experiments using the eir1-4 PIN2::PIN2:Ub:Venus lines? Also, now new lines are introduced such as eir1-4 PIN2::PIN2:mCherry and eir1-4 PIN2::PIN2:Cherry (?) that are used to demonstrate the dark response (PIN2 degradation). Here an effect of exogenous BL was again noted. However, the response to BL of dark-grown roots is quite different to what is normally observed in the light, so what was observed here could be a totally different effect. At least longer-term root growth experiments should be included here to ensure the effects are really caused by enhanced BR signaling.
5. In Fig.4 experiments are presented to show a direct effect of BL application on the distribution between PM and cytoplasm of PIN2:Venus. While I fail to see very much difference between 4a and b, the plots in c suggest a statistically significant effect between the amount of PIN2:GFP in the PM and in the cytoplasm. However, it now turns out according to the description of Fig.4c that here the PIN2::PIN2:Ub lines are used for one analysis (left hand plot) and the PIN2::PIN2 harboring lines in the right-hand plot. I think this is quite misleading to the reader if the authors imply that the effect seen is due to the presence of the bri1-6 allele only.
6. In the last section experiments are presented that are aimed to show an effect of BR signaling on root curvature and growth. In Fig.5a and b effects are reported that suggest that eir1-4 roots are indeed slightly more sensitive at lower concentrations of BL although I would be careful to draw that conclusion, to me it seems that since eir1-4 roots are shorter anyway, it is hard to distinguish these two effects. A similar problem arises when the gravitropic responses are compared, the effect between eir1-4 appears to override any potential effect of exogenous BL. Unfortunately, in my copy of the ms there was nothing to be seen in fig.6d-g. A final comment on the results presented in this Figure is that it should be explicitly stated how the results were actually obtained, the Methods only refers to ImageJ which in my experience is a simplified version of software used for digitized imaging.
7. In Figure 7, again new constructs are introduced that are now referred to as PIN2p::PIN2:Venus

(?!). Using the more sensitive R2D2 reporters I am willing to accept that there is a mild effect of BL application on the gravitropic responses, but I fail to see the effect linked with redistribution of the PIN2p::PIN2:Venus protein, but perhaps with enhanced production?

8. The last part of the ms deals with a mathematical model aimed at evaluating the effects of BL-stimulated redistribution of PIN2 on the gravitropic response. The results suggest that the effects of PIN2 redistribution on auxin steady-state are minor and in line with the reported data. I fail to see how the authors can then conclude that variations in abundance and localization of PIN2 could then be an efficient means to fine-tune the responses.

Reviewer #3 (Remarks to the Author):

In the first part of their paper, the authors show that brassinolide signalling inhibits PIN2 endocytosis, thereby enhancing PIN2 levels at the plasma membrane. This part of the paper is convincing, with experimental data strongly supporting their interpretations.

Next the authors move on to investigate the role of PIN2 and the impact of brassinolide signalling on it in gravitropism. The authors conclude here that (endogenous) brassinolides contribute to PIN2 asymmetry, and that PIN2 asymmetry subsequently reduces rather than enhances gravitropic auxin asymmetry. Here I strongly feel that conclusions are not sufficiently backed by the experimental data and especially the computational modeling results are in their current form non-convincing.

In more detail:

The authors show that upon eBL addition

- auxin asymmetry is enhanced (Fig7h)
- gravitropic bending proceeds more rapidly (Fig6h)
- gravitropic bending overshoots (Fig6h)
- yet PIN2 asymmetry is (seemingly) absent

Combined this seems to suggest that PIN2 asymmetry counteracts rather than contributes to auxin asymmetry and gravitropic bending and this the authors then confirm with modeling (more on this later). Obviously, if this is indeed true, this would be a very interesting and exciting finding, yet to make such a strong claim also strong evidence in its favor is required.

To do so it is important to consider and test alternative explanations.

I could conceive at least 3 alternative explanations for why eBL causes an enhanced gravitropic response and auxin asymmetry yet appears to reduce PIN2 asymmetry:

1) Since eBL inhibits PIN2 endocytosis it enhances PIN2 levels, causing PIN2 levels at both upper and lower sides to be in the upper fluorescence range. Perhaps there is an asymmetry but it can not be picked up? Notably, the authors show only upper versus lower ratios, but no absolute levels so it is hard to assess the likelihood of this scenario.

2) Perhaps PIN2 asymmetry acts as a signal, and overall PIN2 levels, since PIN2 is responsible for transport to the elongation zone, as an amplifier. So perhaps the eBL induced reduction of PIN2 asymmetry is overcompensated by the concomitant eBL induced increase of overall PIN2 levels. (Note that in the above argument I assume that higher auxin arises at the higher PIN2 side, something for which the authors in their model show the opposite, I will turn to this shortly.) Thus, in the model a fairer comparison between control and +eBL would be the following simulation set:
control: substantial PIN2 asymmetry with control PIN2 levels

+eBL: limited PIN2 asymmetry with elevated rather than normal PIN2 levels

3) Possibly the eBL effect on reducing endocytosis is not PIN2 specific, and also AUX1 membrane levels increase. Combined with the importance of auxin induced AUX1 expression for auxin asymmetry, this may lead to an amplification of AUX1 and hence auxin levels at the lower side. Despite the fact that AUX1 has repeatedly been demonstrated to be of critical importance for tropisms, the authors have not investigated what happens for AUX1 under eBL application.

Importantly, the fact that eBL, which reduces the PIN2 degradation that has by others been demonstrated to be essential for termination of the PIN2 and auxin asymmetry, causes gravitropic overshooting, seems to support a promoting rather than the here acclaimed diminishing effect of PIN2 asymmetry on tropic responses.

The authors support for their interpretation of PIN2 asymmetry diminishing auxin asymmetry currently relies solely on their computational model. Unfortunately, there are several serious problems with this model, precluding that we currently accept its outcomes at face value.

The first and foremost problem lies in the interplay of what they investigate -PIN2 asymmetry- with the boundary condition of connecting to a virtual shoot. Of course, a connection to a virtual shoot with influx coming from this shoot into the vasculature and efflux from the outer tissues to the shoot should be incorporated in these types of models. However, in the current model, it is this virtual shoot connection that is probably generating the results. Put simply, if you make the virtual shoot connection directly next to the elongation zone in which the PIN2 asymmetry is applied you simply get that more PIN2 causes more export to the virtual shoot and hence less auxin, and hence a reversal between PIN2 and auxin asymmetry when PIN2 asymmetry is simulated in isolation and the repressive effect of PIN2 asymmetry on auxin asymmetry when simulating gravitropism. This is in my opinion an artifact of the model implementation, not a genuine result! It is in this context important to note that in the model the auxin asymmetry in the lateral root cap, which has no shoot connection and hence not this auxin loss, is aligned with the PIN2 asymmetry: more auxin where there is more PIN2 (Fig 8b)!

The severity of the auxin loss and its interference with the PIN2 asymmetry under investigation can be easily avoided by incorporating also a differentiation zone in the model, in which PIN2 levels taper off and PIN patterns become more lateralized -thereby significantly reducing severity of auxin loss- and also no PIN2 asymmetry occurs under gravitropism (see eg. Figure 1 in Baster et al, 2013 Embo Journal) -thus separating the boundary condition from the subject of study- and then have the virtual shoot connection attached to this more shootward differentiation zone.

The auxin loss problem is further aggravated by the specific model settings used by the authors, who in contrast to other recent studies do not incorporate auxin induced AUX1 expression that can help hold on to auxin if there is more auxin, and also do not incorporate laterally inward oriented PIN2 in the epidermis and cortex enabling reflux of auxin into the vasculature. As a consequence in the model used here there will be a fairly strong positive correlation between auxin loss and PIN2 level.

I strongly suspect that if the authors include an extended differentiation zone with tapering off, symmetric PIN2 levels, laterally oriented PIN2, and auxin dependence of AUX1 expression that were both recently shown to be critical for root tropisms, auxin and PIN2 asymmetry will also be aligned in the elongation zone and PIN2 asymmetry will contribute to rather than diminish auxin asymmetry. Notably, this would require an alternative explanation for why eBL enhances gravitropic responses while reducing PIN2 asymmetry.

Note also that the squarish root topology used by the authors has been abandoned by most research groups in the field in favor of a realistic root tip layout, and that this has been demonstrated to impact the auxin patterns obtained for simulated tropisms.

Finally, it seems that the authors apply a single polar orientation for cortical PIN2 yet data show a flip in cortical PIN2 polarity from meristem to elongation zone. Again, this has been shown to be important for gravitropism.

In conclusion, I can not recommend the paper in its current form for publication and recommend that particularly the modeling needs to be substantially improved.

Reviewer #4 (Remarks to the Author):

The hormonal regulation of plant growth and development represents a major focus by Plant Science researchers. The Arabidopsis root has provided a popular experimental system where many of the most important mechanistic insights have arisen. The current manuscript by Retzer et al, reports how another important class of plant hormone termed brassinosteroids (BR) regulates gravitropism via the auxin efflux carrier PIN2. This represents a significant finding as this important root adaptive response is reported to be primarily controlled via auxin.

The manuscript initially describes how the authors screened for signals controlling endocytosis of ubiquitinated PIN2 (a focus of the lab) leading to the discovery that BR plays a key role. Through a series of elegant experiments using novel tools, the authors convincingly demonstrated that BR caused PIN2 stabilisation at the plasma membrane (PM). By monitoring a YFP tagged ubiquitinated PIN2 form in the background of several BR synthesis and signalling mutants or equivalent chemical inhibitors, the authors observed that the BR transduction pathway controls trafficking of this key root auxin transport protein. They went on to demonstrate that this does not reflect BR regulated changes in PIN2 gene expression, but that BR impacts this ubiquitinated protein's endocytotic sorting via the TOL/ESCRT machinery.

As I read and digested the significance of the results described in the manuscript, I wanted to understand how this may impact the role of PIN2 during root gravitropism, as a clear function for BR has not been established with this adaptive response to date. I was therefore delighted that the authors next considered the wider adaptive importance of this BR-auxin crosstalk mechanism by studying the impact of BR treatment and/or mutants on root gravitropic bending kinetics. Intriguingly, they observed that BR treatment conferred a hyper-responsive behaviour, as wild type (WT) roots (but not *pin2*) exhibited over-bending after a 90 degree gravity stimulus.

To determine which phase(s) of the root gravitropic response BR regulation of PIN2 may be important, the authors characterised the behaviour of PIN2-VENUS reporter* during a root gravitropic response. They observed that PIN2-VENUS undergoes elevated levels on the lower side of gravity stimulated roots (after 90'), which was disrupted in BR treated roots.

*Did they also monitor the effect of gravistimulation on the PIN2-Ub-VENUS reporter?

To understand the impact that this transient up-regulation of PIN2 on the lower root side may have on auxin levels, the authors employed a previously published mathematical model. The model revealed that this regulatory mechanism served to counter-act the earlier formation (within minutes) of the lateral auxin gradient. Hence, the BR-PIN2 mechanism appears to function during the last phase of a gravitropic response, serving to attenuate the lateral auxin gradient, stopping the root over-bending. This point could be spelt out much clearer in the discussion (which currently under sells the significance of the authors findings). To date, little is known about this last phase of a gravitropic response other than gravity-sensing statolith's rolling back into position as a root reaches its mid-point in bending.

I was left wondering after reading the manuscript, so where does BR signalling fit into a root gravitropic response? It's clear from the authors use of the BR response reporter BES1-GFP that there is a transient BR response on the lower side of roots 90' after a gravity response, coinciding with elevated PIN2 levels, but how is the BR response induced? In fairness, this is beyond the scope of the current manuscript, but it's a question I hope the authors or readers of this manuscript will follow up in the future.

Response to the reviewers

Reviewer #1

Previous studies, including works from the Luschnig lab, demonstrated that dynamic sorting of PIN2 between subcellular compartments, in the upper and the lower part of the bending root guides differential distribution of auxin and hence differential growth in response to gravistimulation. The level of the PIN2 protein is also modulated by brassinosteroid with no apparent effect on transcription. Few studies also demonstrated that high BR enhances root response to gravity.

Retzer et al provide new insight into the mechanism by which BR modulates PIN2 and propose an alternative model for the role of PIN2 in gravitropism. The authors took advantage of a transgenic line (*eir1-4 PIN2::PIN2:ub:Venus*) where PIN2 is constitutively ubiquitinated and thus primarily sorted for vacuolar degradation. Using this line, the authors convincingly demonstrated that BR and its canonical pathway (BRI1/BIN2) prevents vacuolar targeting and degradation of PIN2. Interestingly, it was also demonstrated that BL overrides the differential accumulation of PIN2, but the auxin gradient (as shown by the DII Venus marker) and root bending is still maintained. The authors therefore concluded that PIN2 is not necessary for gravitropic root bending. Assisted by a simulation for auxin transport, the authors propose an alternative model, where the role of differential PIN2 abundance is to counteract root bending.

Overall, this is an interesting study on one of the fundamental plant responses. However, the following aspects of this work remained somewhat unclear:

Response: Thank you for these encouraging lines!

1. The authors performed several experiments, demonstrating that BL enhances PIN2 abundance in the plasma membrane by counteracting the constitutive vacuolar targeting and degradation of PIN2. Thus, exposure to BL is functionally equivalent to loss of PIN2 ubiquitylation, as in the *pin212K-R*, a construct previously made and characterized by the authors using similar experiments (PNAS, 2012). *pin212K-R*, an active transporter, failed to complement the *pin2* mutant in physiological assays (PNAS, 2012). I therefore wonder: does this mutation and high BL have a similar effect on gravitropism? (also, is *pin212K-R* levels insensitive to BRZ?).

Response: thank you for pointing this out! Indeed, when we observed eBL effects on PIN2 sorting and root gravitropism, we considered the *pin2K-R* alleles for testing PIN2 function. Leitner et al. (2012; <https://doi.org/10.1073/pnas.1200824109>) demonstrated defects in ubiquitylation and vacuolar targeting of *pin2K-R* alleles, apparently causing increased retention at the plasma membrane. Such plasma membrane retention resembles brassinolide effects on sorting of PIN2, and we therefore tested eBL effects on our published *eir1-4 pin2^{12K-R}* and *eir1-4 pin2^{17K-R}* lines. However, neither did root gravitropism of *pin2K-R* lines resemble that of wild PIN2 lines incubated in presence of eBL, nor did we observe pronounced rescue of *pin2K-R* agravitropic root growth in response to eBL treatment. IAA-retention assays indicated functionality of *pin2K-R* in cellular auxin efflux when expressed in BY-2 cells (Leitner et al., 2012). However, these experiments did tell, whether or not functionality of *pin2K-R* alleles matches that of wild type PIN2. We addressed this issue by testing *pin2K-R* seedling growth in presence of ACC (a precursor of ethylene). Since *eir1-4* roots are less responsive to ACC due to altered auxin transport, we used this growth assay as a morphological read-out for the activity of *pin2K-R* alleles. Notably, root growth of all tested *pin212K-R* and *pin217K-R* lines was found to exhibit a partial resistance to ACC, intermediate between *eir1-4 PIN2::PIN2* and *eir1-4* mutant lines, suggesting that functionality of the *pin2K-R* alleles differs from that of wild type PIN2. This could explain the differing brassinolide responses of seedlings expressing either wild type PIN2 or *pin2K-R* lines, and prevented us from further using these mutant lines for analysis of brassinolide effects on PIN2 sorting and root gravitropism.

2. Regarding the model: The auxin gradient of gravistimulated root is maintained in the presence of high BL, despite similar PIN2 sorting dynamics. Can the authors exclude the possibility that the enhanced curvature is simply due to the interaction between BL and auxin activity on target genes that further promote cell elongation in the upper side?

Response: Important point! Given the various responses to BL in the control of root development, it seems impossible to categorically exclude such a mode of action. Results in our m/s demonstrate that eBL treatment affects root curling/waving, even in a *pin2* loss-of-function mutant, indicative of eBL effects on root bending uncoupled from PIN2-mediated transport. These responses could e.g. be linked to brassinolide effects on the cytoskeleton as indicated by Lanza and colleagues (2012 Dev. Cell; <https://doi.org/10.1016/j.devcel.2012.04.008>). On the other hand, eBL treatment of *pin2* mutants is insufficient to restore root gravitropism, supporting long-

standing evidence, which demonstrated that PIN2-mediated auxin transport is indispensable for correct interpretation of a gravity signal - even upon increasing eBL levels.

The revised version of our m/s now includes results from additional experiments, in which we further tackle crosstalk between auxin and eBL in the control of root bending. eBL treatment of the *tir1-1 afb1-1 afb2-1 afb3-1* quadruple auxin receptor mutant results in increased root curliness but completely fails to restore root gravitropism (Fig. S9c). These responses resemble those of *pin2* mutants, substantiating a critical role for auxin transport and signaling for correct interpretation of brassinolide signals in the control of gravitropic root bending. Furthermore, and as suggested by reviewer #1 (see point 4), we assessed eBL effects on BES1-GFP expression in gravistimulated roots. These experiments demonstrated that exogenously applied eBL overcomes formation of a BES1-GFP signal gradient in gravistimulated roots (Fig. S10). Given that roots still exhibit gravitropic root bending under these growth conditions, we concluded that a brassinolide signaling gradient, acting via differential induction of target genes is not absolutely required for gravity-induced root bending. Nevertheless, alterations in the kinetics of root bending that can be observed in response to brassinolide treatment, supports scenarios, with differential eBL signaling affecting timing and magnitude of gravitropic root growth, presumably acting in conjunction with additional signaling events, impacting on shootward auxin transport.

In another set of experiments, we extended our analysis of gravitropic root growth in response to brassinolide by testing PIN2-Venus and PIN2-ubq-Venus lines. *eir1-4 PIN2p::PIN2:VEN* shows an enhanced responsiveness to gravistimulation, similar to wild type roots (Fig. S9b). Strikingly, we also observed a partial restoration of root gravitropism of *eir1-4 PIN2p::PIN2:ubq:VEN* seedlings in response to brassinolide, which we did not observe with *eir1-4* controls (Fig. S9b). eBL-induced stabilization of an otherwise constitutively degraded allele of PIN2, thus appears sufficient for at least partial restoration of gravitropic root growth. These observations are in agreement with our working hypothesis, in which stabilization of PIN2 in response to eBL does not abolish gravitropic root bending.

In the revised version of the m/s, we carefully edited and adapted our conclusions, emphasizing the various effects of eBL on root growth, with brassinolide effects on PIN2 distribution contributing to the control of gravitropic root bending.

3. After hours of incubation with BL, cell elongation starts closer to the root tip (see also images in Figure 7h, of note: how long the roots were pre-treated with BL?). Will the simulation be affected by taking into consideration the distance from the root tip?

Response: We agree that, as the morphology of BL-treated roots is different from control roots, it could affect the outcome of the model. Therefore, we have tested different geometries of the root in our simulations (Fig S11; eBL-treated geometry presented in Fig. 11e), and found that differences in cell elongation do not affect the major conclusion of our model: auxin ratio is inversely proportional to the ratio of PIN2 on upper and lower sides of the root meristem ($IAA_{lower/upper} \sim PIN2_{upper/lower}$). In other words, there is less auxin retained in the cells, if there is more PIN2 at the plasma membrane and *vice versa*, more auxin inside the cells if there is less PIN2 present. The quantitative differences between simulated eBL-treated root geometry and control root are presented in Fig. S11g: The ratio between $IAA_{lower/upper}$ side vs. $PIN2_{upper/lower}$ side differs by a maximum of 0.5% in case of no gravistimulation and by a maximum of 5% in case of gravistimulation.

4. The accumulation of BES1 in the lower side is a very interesting observation. Does exposure to BL (or when in *bri1* mutant) abolish the differential accumulation BES1?

Response: This is another relevant point and we did the requested experiment. Gravistimulation of 35S-BES1-GFP seedlings in presence of eBL in the medium, no longer coincided with formation of a lateral GFP gradient, suggesting that external brassinolide interferes with differential eBL signaling/responses in gravistimulated roots (Fig. S10). Such eBL treatments, however, do not interfere with the root's ability to respond to gravistimulation, but modify the kinetics of gravity-induced root curvature. This argues for a scenario, in which differential brassinolide signaling and its effects on PIN2 distribution and sorting, participate in timing and kinetics of gravitropic root bending.

5. Figure 6h: Over-response to gravistimulus by high BL was previously described in the following missing references: Involvement of Brassinosteroids in the Gravitropic Response of Primary Root of Maize, Kim et al, Plant phys, 2000; Brassinosteroids Stimulate Plant Tropisms through Modulation of Polar Auxin Transport in Brassica and Arabidopsis, Li et al, Plant Cell, 2005.

Response: Both references can be found in the revised m/s.

6. Root elongation of *pin2* is shown here to be more sensitive to inhibition by BR. Could the authors discuss why?

Response: When comparing wild type and *eir1-4* root growth at low eBL concentrations, we observed reduced root elongation of *eir1-4*, whereas at higher eBL concentrations, both lines exhibited comparable responsiveness to hormone treatment (Fig. 5a). Low brassinolide concentrations have been described to promote, rather than inhibit root elongation of wild type Arabidopsis (e.g. Müssig et al., Plant Phys., 2003; [doi/10.1104/pp.103.028662](https://doi.org/10.1104/pp.103.028662); Kim et al., Plant Cell Physiol., 2007; [doi: 10.1111/j.1365-3040.2007.01659.x](https://doi.org/10.1111/j.1365-3040.2007.01659.x)), which we also observed with wild type seedlings. Differences in *eir1-4* and Col0 root growth that we observed in response to low concentrations of eBL, thus, might argue for a role of PIN2 in the control of root growth, under such growth conditions. Perhaps, such low eBL concentrations influence PIN2-mediated shootward auxin transport – thereby promoting root elongation, whilst additional brassinolide effects on e.g. cell wall and cytoskeleton elements might play only subordinate roles under these particular growth conditions. Or vice versa, effects of low eBL concentrations on determinants of root growth, other than PIN2, might be more pronounced in the absence of functional PIN2. With our limited knowledge however, it is difficult to draw any valid conclusions. Therefore, and due to the fact that this m/s focusses on the control of root gravitropism by eBL, we decided to omit this dataset in the revised version of our m/s.

7. Line 80: add Figure S2 d

Response: This has been edited accordingly.

8. Figure 1 and 2 – Indicate the number of roots used for the analysis

Response: This information can now be found in the Figure legends

9. Figure 2h and I - what was the incubation time?

Response: We have added this information

10. Figure 5C: add PIN2::PIN2:Venus

Response: We have edited this section accordingly.

11. Western blot showed no increase in PIN2 levels in response to BL, but in Figure 7B it appears that the levels of PIN2 are enhanced by BL in the plasma membrane (compare to 0 to 150 min).

Response: We agree, these images could indeed give this impression. However, this likely results from variations in expression intensities that can be observed in the large sample of PIN2-Venus individual roots that we analyzed. In contrast, we never observed a corresponding increase of PIN2(-Venus) levels on Western blots, performed with eBL-treated root material.

The misleading impression could be strengthened further by the fact that the optical axis in the images used in original Fig. 7b was not perfectly perpendicular to the root's central plane, resulting in very distorted/blurred signals in the maximum projections on display. We therefore have replaced the original images by maximum projections of roots properly positioned in the optical plane. In addition, we redid the entire analysis of PIN2-Venus kinetics in gravity-responding roots, adding additional timepoints and reducing laser intensities (Fig. 6 in the revised m/s), in order to avoid saturation effects, upon signal quantification.

Reviewer #2 (Remarks to the Author):

1. The manuscript by Retzer et al. describes a comprehensive series of experiments aimed to link PIN2 PM stability mediated by BR signaling to affect root gravitropism. I have a number of comments on the work.

Response: Thank you for comments and suggestions. We tried to address all the points raised by reviewer # 2

2. The construct described as PIN2:Ub:Venus is introduced as being previously reported and characterized by Leitner et al. 2012. In that publication a construct is described that is referred to as PIN2:ubq:VEN and I assume that this is the same construct used here, so it would be good to indicate that and why the name has changed since. But what is somewhat more problematic is that PIN2:ubq:VEN is described as not being able to complement *eir1-4* (Fig.4I and J and description in the text of Leitner et al. 2012). Rightfully so, this was used as a strong argument to underscore the importance of ubiquitination in PIN2 endocytosis. If the constructs are the same, then it should be clearly stated that we are dealing also in the present work with a non-functional reporter in the mutant background. If not, a brief recap should introduce the new construct and state why the authors actually use the *eir1-4* PIN2::PIN2:Ub:Venus background instead of wt in Fig. 2 and thereafter. This may or may not have impact on the interpretation of the experiments in this work, because to me it appears to be a somewhat artificial experimental system.

Response: We apologize for this confusion. We made use of the original nomenclature (introduced by Leitner et al., 2012; <https://doi.org/10.1073/pnas.1200824109>) in the revised version of our m/s. For the functionality of ubiquitylated PIN2, please see our response to point 3.

3. The next series of experiments (Fig.2-Fig.4) deal with demonstrating the involvement of the BR signaling system in the observed stabilization of PIN2:Ub:Venus at the PM. This is all well done and various tests are employed. One control that I suggest including is to determine whether the *eir1-4* PIN2::PIN2:Ub:Venus lines show any change in BL insensitivity compared to wt and the *eir1-4* mutant. This would confirm whether the treatments used already start at another point in the existing steady state of BR signaling.

Response: PIN2:ubq:VEN has originally been described by Leitner et al., (2012; <https://doi.org/10.1073/pnas.1200824109>), as a constitutively degraded version of PIN2. Here we used this line as a tool to screen for effectors stabilizing PIN2. As rightly pointed out by referee #2, interpretation of results could be problematic, when employing a non-functional allele. In the revised version of our m/s we therefore extended our analysis of root gravitropism in response to eBL and added *eir1-4* PIN2p::PIN2:VEN as well as *eir1-4* PIN2p::PIN2:ubq:VEN to our growth assays, as both of these lines have been extensively used for PIN2 sorting analyses, presented in our m/s. When grown on eBL, *eir1-4* PIN2p::PIN2:VEN shows enhanced root curvature similar to the growth responses that we observed with wild type seedlings (Fig. S9b). This would be in line with our findings, indicating that brassinolide-dependent stabilization of PIN2:Venus does not abolish gravitropic root bending. Importantly, we found that eBL treatment partially restores gravitropic root bending of *eir1-4* PIN2p::PIN2:ubq:VEN seedlings as well (Fig. S9b). This indicates that PIN2:ubq:VEN has retained -at least limited- functionality in auxin transport, despite ubq and Venus tags that have been added to the protein. Partial restoration of root gravitropism is only observed upon stabilization of PIN2:ubq:VEN at the plasma membrane, whereas no such response is observed in *eir1-4* PIN2p::PIN2:ubq:VEN on control medium lacking brassinolide as well as in *eir1-4* controls (Fig. S9b). Brassinolide-dependent accumulation of PIN2 at the plasma membrane thus appears sufficient for partial restoration of gravitropic root growth, which corroborates outcome and interpretation of our experiments performed with *eir1-4* PIN2p::PIN2:ubq:VEN.

4. In the section starting at line number 175 it is misleadingly announced that wild type PIN2 is now investigated whereas the text and figures indicate that it is rather PIN2:Venus. Here I am getting somewhat confused by the results presented in Fig.5 and S6, because when using the PIN2::PIN2:Venus construct, again assuming it is the same as reported previously, that is complementing the *eir1-4* mutation there is very little effect to be seen after application of BL. So how meaningful are all of the reported experiments using the *eir1-4* PIN2::PIN2:Ub:Venus lines? Also, now new lines are introduced such as *eir1-4* PIN2::PIN2:mCherry and *eir1-4* PIN2::PIN2:Cherry (!) that are used to demonstrate the dark response (PIN2 degradation). Here an effect of exogenous BL was again noted. However, the response to BL of dark-grown roots is quite different to what is normally observed in the light, so what was observed here could be a totally different effect. At least longer-term root growth experiments should be included here to ensure the effects are really caused by enhanced BR signaling.

Response: Thank you very much for noticing! We rephrased this section in order clarify this misleading statement. Now we explicitly stress the fact that tagged PIN2 has been used for most of the experiments, and point out that expression of these reporter constructs rescue a *pin2* null allele.

We are sorry that this section has caused confusion. In fact, we tried to highlight the fact that PIN2-Venus levels show only limited responsiveness to brassinolide, when compared to constitutively ubiquitinated and degraded PIN2-Venus. Nevertheless, our experiments in which we tested endocytic sorting and degradation of PIN2-Venus/mCherry, demonstrate a stabilizing brassinolide effect, similar to its effects on PIN2:ubq:VEN. This includes the analysis of NAA effects on the stability of PIN2:Venus as well as vacuolar accumulation of PIN2:mCherry (Fig. 4g-l). This is in agreement with a scenario, in which brassinolide preferably antagonizes endocytic sorting of the PIN2 pool, bound for vacuolar degradation.

We used the mCherry tag for vacuolar accumulation assays, because it is less pH-sensitive when compared to the Venus tag. This facilitates quantification of reporter signals in acidic compartments. We indicate this fact in the revised version of our m/s (methods section). We have also added long-term eBL treatments of *eir1-4 PIN2p::PIN2:mCherry*, incubating the seedlings for 16 hours, which is presented in Fig. S7j-l.

Thank you very much for pointing out these ambiguities! PIN2:Cherry now reads PIN2:mCherry throughout the entire revised manuscript.

5. In Fig.4 experiments are presented to show a direct effect of BL application on the distribution between PM and cytoplasm of PIN2:Venus. While I fail to see very much difference between 4a and b, the plots in c suggest a statistically significant effect between the amount of PIN2:GFP in the PM and in the cytoplasm. However, it now turns out according to the description of Fig.4c that here the PIN2::PIN2:Ub lines are used for one analysis (left hand plot) and the PIN2::PIN2 harboring lines in the right-hand plot. I think this is quite misleading to the reader if the authors imply that the effect seen is due to the presence of the *bri1-6* allele only.

Response: Apparently, this was a bit mis-leading in our original manuscript. As indicated in the Figure legend, all panels presented in Fig. 4 of the original manuscript, describe distribution, sorting and protein levels of PIN2:ubq:VEN. In the revised Fig. 3, we have tried to clarify this point, by editing the figure legend and by adding the reporter line identifier (*eir1-4 PIN2::PIN2:ubq:VEN*) in all the graphs that come with the image panels.

6. In the last section experiments are presented that are aimed to show an effect of BR signaling on root curvature and growth. In Fig.5a and b effects are reported that suggest that *eir1-4* roots are indeed slightly more sensitive at lower concentrations of BL although I would be careful to draw that conclusion, to me it seems that since *eir1-4* roots are shorter anyway, it is hard to distinguish these two effects. A similar problem arises when the gravitropic responses are compared, the effect between *eir1-4* appears to override any potential effect of exogenous BL. Unfortunately, in my copy of the ms there was nothing to be seen in fig.6d-g. A final comment on the results presented in this Figure is that it should be explicitly stated how the results were actually obtained, the Methods only refers to ImageJ which in my experience is a simplified version of software used for digitized imaging.

Response: We fully agree with this point! A slight alteration in responsiveness to low eBL concentrations will not necessarily indicate that *eir1-4* is hypersensitive to eBL. Reports indicated that wild type roots exhibit increased root elongation in response to low eBL concentrations (e.g. Müssig et al., Plant Phys., 2003; [doi:10.1104/pp.103.028662](https://doi.org/10.1104/pp.103.028662); Kim et al., Plant Cell Physiol., 2007; [doi: 10.1111/j.1365-3040.2007.01659.x](https://doi.org/10.1111/j.1365-3040.2007.01659.x)). Differences in *eir1-4* and wild type root growth that we observed in response to such low eBL concentrations, thus, could argue for a role of PIN2 in the control of root growth, specifically under such growth conditions. This, together with the fact that brassinolide impacts on root growth via various pathways, makes it difficult to interpret this finding. Therefore, and since our m/s almost exclusively addresses crosstalk between eBL and PIN2 in the control of root gravitropism, we decided to omit this Figure panel in the revised version of the m/s. We are sorry for the problems, visualizing Fig.6d-g of our original manuscript. These images show directional root growth of Col0 and *eir1-4* roots in response to eBL. We do hope that these panels do not cause any further problems in the revised version of the m/s (now Fig. 5b-e).

As rightly pointed out by referee 2, eBL exerts strong effects on waving and bending of Arabidopsis primary roots, which we observed even in the *eir1-4/pin2* null allele. This is indicated by increased *eir1-4* root curling upon eBL treatment (Fig. S9a) as well as by the induction of *eir1-4* root curving upon gravistimulation in the presence of eBL (Fig. 5i; Fig. S9b). Brassinolide-induced adjustments in the frequency of root waving/curving in *eir1-4*, thus are likely not caused by alterations in PIN2-controlled shootward auxin transport from the root tip into the elongation. Instead, these effects could be explained by responses, different from PIN2-dependent auxin transport; e.g. hormone effects on cytoskeleton function, as suggested by Lanza and colleagues (2012; Dev. Cell;

<https://doi.org/10.1016/j.devcel.2012.04.008>). Apart from that, our experiments demonstrated that, whilst inducing root curving, brassinolide treatment does not rescue gravitropic root bending (i.e. reorientation of root growth in the direction of the gravity vector) in the *eir1-4* loss-of-function mutant. PIN2-dependent auxin transport thus, seemingly contributes to the control of gravitropic root growth in response to brassinolide. This is underlined further by experiments added to the revised version of the m/s, indicating that stabilization of constitutively ubiquitylated PIN2 by brassinolide treatment is sufficient to partially restore directional root growth in response to gravity (Fig. S9b). The apparent correlation between stabilization of PIN at the plasma membrane (as independently demonstrated for PIN2-Venus and PIN2:ub-Venus) and enhanced gravitropic root bending, are in line with the outcome of our simulation approaches, summarized in Fig. 8 and Figs. S11-S15 of our revised manuscript. Nevertheless, whilst PIN2-mediated auxin transport seemingly contributes to eBL effects on root growth, combinatorial brassinolide effects on cytoskeleton, cell wall, auxin transport, etc., are likely to define the kinetics of root bending. We emphasized the occurrence of such combinatorial effects in our revised m/s.

We have added the requested information on data acquisition in the Methods section of the revised version of our m/s.

7. In Figure 7, again new constructs are introduced that are now referred to as PIN2p::PIN2:Venus (?!). Using the more sensitive R2D2 reporters I am willing to accept that there is a mild effect of BL application on the gravitropic responses, but I fail to see the effect linked with redistribution of the PIN2p::PIN2:Venus protein, but perhaps with enhanced production?

Response: Thanks for pointing this out! We completely redid our quantification of PIN2-Venus distribution in gravistimulated roots, using reduced laser intensities upon imaging (to avoid saturation effects), and including additional time points for the assessment of PIN2-Venus kinetics (Fig. 6, in the revised m/s). Results from these experiments are consistent with those from the original manuscript, demonstrating reversible PIN2-Venus gradient formation under control conditions, whereas no such gradient can be observed in the presence of brassinolide. We also replaced the original PIN2-Venus images, now using images from the repetition of our quantification, with the transversal cell walls being perfectly perpendicular to the roots central plane. Gradient formation under control conditions and effects of brassinolide on such gradient formation can be readily observed in these images. We hope that these modifications will help to resolving this issue.

8. The last part of the ms deals with a mathematical model aimed at evaluating the effects of BL-stimulated redistribution of PIN2 on the gravitropic response. The results suggest that the effects of PIN2 redistribution on auxin steady-state are minor and in line with the reported data. I fail to see how the authors can then conclude that variations in abundance and localization of PIN2 could then be an efficient means to fine-tune the responses.

Response: We thank reviewer #2 for noticing this seemingly logical pitfall in the text. It arises, because the effects of PIN2 asymmetry on auxin ratios are indeed smaller than the effects of PIN3/PIN7 asymmetry. However, for fine-tuning directional root growth, it is not required to completely annihilate the auxin asymmetry established by initial PIN3/PIN7 asymmetry (as a full counteraction would stop gravitropic bending altogether). Rather we were seeking to explain the differences between auxin ratios measured in eBL-treated roots (median auxin ratio = 2.3) and in control roots (median auxin ratio = 1.67).

We therefore have deleted the following misleading sentence: 'It appears from our analysis, that variations in PIN2 abundance do have minor consequences for auxin distribution, when compared to the input of PINs in root cap cells. In agreement, we observed only a small increase in differential auxin signaling in gravistimulated root meristems, upon interference with PIN2 gradient formation (Fig. 7h)'. We rephrased this section to make our conclusions clearer and unambiguous.

Reviewer #3 (Remarks to the Author):

In the first part of their paper, the authors show that brassinolide signalling inhibits PIN2 endocytosis, thereby enhancing PIN2 levels at the plasma membrane. This part of the paper is convincing, with experimental data strongly supporting their interpretations.

Next the authors move on to investigate the role of PIN2 and the impact of brassinolide signalling on it in gravitropism. The authors conclude here that (endogenous) brassinolides contribute to PIN2 asymmetry, and that PIN2 asymmetry subsequently reduces rather than enhances gravitropic auxin asymmetry. Here I strongly feel

that conclusions are not sufficiently backed by the experimental data and especially the computational modeling results are in their current form non-convincing.

In more detail:

The authors show that upon eBL addition

- auxin asymmetry is enhanced (Fig7h)
- gravitropic bending proceeds more rapidly (Fig6h)
- gravitropic bending overshoots (Fig6h)
- yet PIN2 asymmetry is (seemingly) absent

Combined this seems to suggest that PIN2 asymmetry counteracts rather than contributes to auxin asymmetry and gravitropic bending and this the authors than confirm with modeling (more on this later). Obviously, if this is indeed true, this would be a very interesting and exciting finding, yet to make such a strong claim also strong evidence in its favor is required.

Response: Thank you very much, for your thoughtful comments and suggestions. We carefully went through the points raised by reviewer #3, with our point-by-point response to be found below.

To do so it is important to consider and test alternative explanations.

I could conceive at least 3 alternative explanations for why eBL causes an enhanced gravitropic response and auxin asymmetry yet appears to reduce PIN2 asymmetry:

1) Since eBL inhibits PIN2 endocytosis it enhances PIN2 levels, causing PIN2 levels at both upper and lower sides to be in the upper fluorescence range. Perhaps there is an asymmetry but it can not be picked up? Notably, the authors show only upper versus lower ratios, but no absolute levels so it is hard to assess the likelihood of this scenario.

Response: We did some extensive editing, and incorporated additional experiments and simulations in the revised version, in which we tried to address the reviewer's concerns.

We redid and extended our analysis of PIN2-Venus signal gradient formation in response to brassinolide. Firstly, we used drastically reduced laser intensities, in order to avoid saturation effects upon signal quantification. Apart from that, we added additional timepoints to our analysis of PIN2 abundance in gravistimulated roots. These experiments confirmed lateral PIN2-Venus signal gradient formation after 90 and 150 minutes, which was not detectable at other time points tested (30, 240, 300 and 400 minutes of gravistimulation). This underlines formation of a transient PIN2 gradient during intermediate-to-late stages of gravitropic root bending. However, we did not observe a comparable signal gradient in the presence of brassinolide, demonstrating that such treatment interferes with the kinetics of gradient formation, as observed in wild type.

We cannot categorically exclude formation of a PIN2 expression or activity gradient, occurring in presence of brassinolide (which is also indicated in the revised m/s). Nevertheless, it appears that in presence of elevated brassinolide levels and at time points investigated, no PIN2 gradient, comparable to control conditions can be detected. From that we concluded that formation of a PIN2 gradient, exhibiting a kinetics as observed in gravistimulated wild type or *eir1-4 PIN2p::PIN2:VEN* roots is dispensable for i) auxin gradient formation – as judged from R2D2 expression analysis, and ii) gravitropic root bending – as indicated by root bending assays.

Baster et al. (*EMBO J.*, 2013; [doi: 10.1038/emboj.2012.310](https://doi.org/10.1038/emboj.2012.310)), provided absolute values for PIN2-reporter signals in gravistimulated roots. We repeatedly tried to obtain absolute values for PIN2 as well (e. g. when preparing datasets used for Abas et al., 2006, *Nature Cell Biology* 8, 249–256; Leitner et al., 2012, *PNAS*, <https://doi.org/10.1073/pnas.1200824109>; and for the current manuscript), but we never succeeded with our approaches. This is mainly the result of variations in PIN2:GFP (used for earlier work) and PIN2:Venus (used in this manuscript) signal intensities, when comparing larger sets of individual roots, even when using low laser intensities. We therefore limited our results to relative signal intensities.

Experimental evidence for a role of PIN2 sorting in mediating brassinolide effects on gravitropic root bending came from the analysis of PIN2-Venus distribution and gravitropic root bending of wild type seedlings. In the revised version of our manuscript, we extended this analysis, testing eBL effects on *eir1-4 PIN2-Venus* reporter lines as well. Consistent with a role for PIN2 in mediating brassinolide effects on root growth, gravitropic root bending of *eir1-4 PIN2::PIN2:VEN* in presence of brassinolide exhibited a kinetics similar to eBL-treated wild type roots (Fig. S9b). Furthermore, brassinolide treatment also resulted in partial restoration of root gravitropism in *eir1-4 PIN2p::PIN2:ubq:VEN*, which was not observed in *eir1-4* control seedlings. In contrast, under regular growth conditions, *eir1-4 PIN2p::PIN2:ubq:VEN* exhibits agravitropic root growth, indistinguishable from *eir1-4*

controls (Fig. S9b). Thus, stabilization of PIN2 at the plasma membrane as is the case in in *eir1-4 PIN2p::PIN2:ubq:VEN* in response to brassinolide, appears to contribute to restoration of differential auxin transport into the root elongation zone, which would be consistent with the outcome of our modelling approaches (see below).

2) Perhaps PIN2 asymmetry acts as a signal, and overall PIN2 levels, since PIN2 is responsible for transport to the elongation zone, as an amplifier. So perhaps the eBL induced reduction of PIN2 asymmetry is overcompensated by the concomitant eBL induced increase of overall PIN2 levels. (Note that in the above argument I assume that higher auxin arises at the higher PIN2 side, something for which the authors in their model show the opposite, I will turn to this shortly.) Thus, in the model a fairer comparison between control and +eBL would be the following simulation set:

control: substantial PIN2 asymmetry with control PIN2 levels

+eBL: limited PIN2 asymmetry with elevated rather than normal PIN2 levels

Response: We thank reviewer for pointing out the importance of the overall PIN2 levels that could i) influence auxin ratios, independently of PIN2 asymmetry, and ii) influence effects of the PIN2 asymmetry.

In the revised manuscript, we now analyze the dependence of auxin ratios on initial PIN2 levels (manifested in our model as PIN2 permeability, see Fig. 8h, Fig.S14a), Importantly, when introducing PIN2 asymmetry, we consistently observed higher auxin concentrations arising at the side with lower PIN2, regardless of initial PIN2 levels used for the simulation. Thus, the main conclusion of our simulations is not affected by the overall PIN2 level: the auxin ratio is always inversely proportional to the ratio of PIN2 on the upper and lower sides of roots ($IAA_{lower/upper} \sim PIN2_{upper/lower}$).

As suggested by reviewer #3, we performed the following simulations:

1) Control root simulated, using our "main model" geometry, with $\bar{P}_{PIN2}^{initial} = 0.5$, $\bar{P}_{AUX1} = 0.7$, $k = 0.81$. Note, that for simplicity we present normalized values of PIN2 and AUX1 permeabilities, divided by "default" values $P_{PIN2} = 0.5 \mu m/s$ and $P_{AUX1} = 0.35 \mu m/s$. Thus, $\bar{P}_{PIN2}^{initial} = 1$ and $\bar{P}_{AUX1} = 1$ correspond to these default values (for literature referring to these values, see Table S1). PIN2 asymmetry measured in control roots corresponds to our model parameter $k = \frac{P_{PIN2}^{upper}}{P_{PIN2}^{lower}} = 0.81$.

The resulting auxin ratio for these parameters $\frac{IAA(lower\ side)}{IAA(upper\ side)} = 1.62$ - median measured in control roots.

(see Fig. S15d, first bar).

2) eBL-treated roots, simulated using the eBL-treated geometry (Fig. S11e) and 2-fold overall PIN2 levels $\bar{P}_{PIN2}^{initial} = 1$. Resulting auxin ratios are presented in Fig. S15d. We tested different levels of PIN2 asymmetry: as in controls ($k = 0.81$, bar #2), limited PIN2 asymmetry ($k = 0.9$, bar#4) and no asymmetry ($k = 1$, bar#7).

We found that indeed, an overall increased PIN2 slightly increases the auxin ratio. However, PIN2 asymmetry loss, even when only partial ($k = 0.9$, bar#4), has a much more pronounced effect on the auxin ratio. Importantly, even a 10-fold increase in overall PIN2 levels in this example, was insufficient to reach the auxin ratio measured under eBL treatment, upon simulating conditions, without PIN2 asymmetry (Fig. S15d, bar #11).

We further explored a wide range of PIN2 permeability values (and also AUX1 levels, see our response to reviewer's point #3, below), and show nonlinearity of its effect on auxin ratio, which we discuss in the revised manuscript and extensively in our revised Supplementary information. Graphs depicting the dependence of auxin ratios on PIN2 levels are presented on Fig. 8h. The auxin ratio dependence on PIN2 asymmetry for different PIN2 values is presented in Fig. S14a and shows that for physiologically relevant PIN2 levels ($0.1 < \bar{P}_{PIN2}^{initial} < 10$), PIN2 asymmetry has a pronounced effect on the auxin ratio. From this analysis we concluded that PIN2 asymmetry will substantially contribute to the dampening of auxin ratios that we observed in controls, exhibiting asymmetric PIN2 distribution. Thus, its effect cannot be neglected when comparing auxin ratios in control vs. eBL-treated roots.

3) Possibly the eBL effect on reducing endocytosis is not PIN2 specific, and also AUX1 membrane levels increase. Combined with the importance of auxin induced AUX1 expression for auxin asymmetry, this may lead to an amplification of AUX1 and hence auxin levels at the lower side. Despite the fact that AUX1 has repeatedly been demonstrated to be of critical importance for tropisms, the authors have not investigated what happens for AUX1 under eBL application.

Response: Following the reviewer's suggestion, we explored the effect of overall AUX1 levels on auxin ratios in gravistimulated roots (Fig. S15a,b,c) (note, that we did not observe an obvious increase in AUX1-YFP signals in response to eBL treatment (Fig. S8), and failed to detect an AUX1-YFP signal gradient in gravistimulated roots, however we cannot rule out changes in AUX1 activity).

In the simulation presented in Fig. S15d, we used a 1.5-fold increase in overall AUX1 levels: $\bar{P}_{AUX1} = 1$, and assumed PIN2 asymmetry in the gravity-responding root. The resulting lower/upper auxin ratio is presented in Fig. S15d, bar #3 and demonstrates that an increase in AUX1 levels, causes an increase in the auxin ratio, similar to the effect of increasing PIN2 levels. Notably, under these conditions, variations in PIN2 asymmetry still act on auxin ratio, quite similar to the results that we obtained upon simulation increasing PIN2 levels. Compare Fig. S15d bar#5 with simulated PIN2 asymmetry, and bar#9, simulating elimination of PIN2 asymmetry: The difference in auxin levels at upper and lower sides of the root is more pronounced in the latter case.

From this analysis we concluded that an increase in AUX1 levels could -to some extent- contribute to the increase in the auxin ratio that we observed in eBL-treated roots. However, loss of PIN2 asymmetry contributes to increased auxin ratios as well, under all conditions tested. Therefore, its effect cannot be neglected, when comparing auxin ratios in control vs. eBL-treated roots. We further demonstrate that upon neglecting PIN2 asymmetry, PIN2 and AUX1 levels need to increase at least 3-fold (see Fig. S15d bar#10) to account for the auxin ratio observed in eBL-treated roots, whereas comparably subtle changes in PIN2 asymmetry can produce comparable effects on auxin ratios.

As pointed out in the revised manuscript, our simulations consider only an overall AUX1 increase. It is conceivable that additional parameters affecting AUX1 function in shootward auxin transport might contribute to differential auxin transport as well. AUX1 activity could for example be affected by local adjustments in pH (e.g., Yang et al., 2006; *Curr. Biol.*, DOI 10.1016/j.cub.2006.04.029; Carrier et al., 2008, *Plant Physiology* <https://doi.org/10.1104/pp.108.122044>). Whilst we cannot exclude the existence of such "reinforcement mechanisms", it will not affect our conclusions about the role of PIN2 asymmetry in directional auxin transport. As consistently demonstrated in all our model setups, higher PIN2 abundance at the plasma membrane will decrease intracellular auxin levels and *vice versa*, regardless of whether this effect is further reinforced by a positive feedback on AUX1 or not.

The main goal of our simulations was to test how the observed PIN2 asymmetry influences auxin distribution in gravity-responding roots. Thanks to the valuable input from reviewer #3, we have extended and refined our simulation of auxin flow in gravity-stimulated roots. Regardless of the parameters being tested, we reproducibly observed diminished auxin gradient formation in response to a gradient in PIN2-mediated auxin flux.

We therefore do suggest that asymmetric PIN2 distribution contributes to a dampening of the lateral auxin gradient, established in the root tip, for the following reasons:

- 1) although the experimentally determined difference in PIN2 abundance at the upper vs. lower side of roots is low ($k = 0.81$), it appears sufficient to contribute about 50% to the difference in auxin ratios, when comparing control and eBL-treated roots.
- 2) When assuming that an overall increase in the levels of PIN2 and/or AUX1 would account for a more pronounced auxin gradient established in response to eBL treatment, then one would expect an at least 3-fold increase in PIN2 and AUX1 levels in eBL-treated roots. No experimental evidence for such differences in protein abundance has been established so far.
- 3) Based on our simulations it appears that adjustments in determinants of auxin transport (i.e. PIN2, AUX1, PIN3/7), contribute to differential auxin flow and gravitropic root bending. We have modified our revised manuscript accordingly. This is also true for diminished PIN2 gradient formation, as observed in response to eBL, which causes a robust enhancement of differential auxin flux under all conditions simulated.

Importantly, the fact that eBL, which reduces the PIN2 degradation that has by others been demonstrated to be essential for termination of the PIN2 and auxin asymmetry, causes gravitropic overshooting, seems to support a promoting rather than the here acclaimed diminishing effect of PIN2 asymmetry on tropic responses.

Response: We agree, such a scenario (i.e. stabilization of PIN2 triggers termination of gravitropic root bending) cannot be excluded, and we postulated related models in some of our earlier work (e.g. Abas et al., 2006, *Nature Cell Biology* <http://doi.org/10.1038/ncb1369>). However, resetting of PIN2 levels to default occurs after around 2- to-3 hours of gravistimulation (see e.g. Abas et al., 2006, *Nature Cell Biology* <http://doi.org/10.1038/ncb1369>;

Leitner et al., 2012, *PNAS*, <https://doi.org/10.1073/pnas.1200824109>; Baster et al. (*EMBO J.*, 2013; doi: [10.1038/emboj.2012.310](https://doi.org/10.1038/emboj.2012.310); this manuscript). At this time point, auxin gradient formation in the columella root cap appears to be no longer driven by asymmetric PIN3/PIN7 distribution (Band et al., 2012; *PNAS*; <http://doi.org/10.1073/pnas.1201498109>). This differs substantially from the situation described in our manuscript, in which PIN2 gradient formation appears to be blocked from the very beginning of gravitropic root bending. This likely results in a different kinetics of auxin flux (also supported by the outcome of our simulation approaches), which makes it difficult to compare auxin distribution/signaling and its consequences on root bending under these differing conditions.

The outcome of our growth assays and simulations of auxin flux argue for a scenario in which it is the PIN2 gradient, counteracting consequences of a steep auxin gradient during mid/late stages of gravitropic root bending, thereby contributing to termination of root bending. Published literature and our simulations of auxin transport in gravity-responding roots (see below) provide ample evidence for participation of additional determinants in the control of differential auxin transport in gravistimulated roots. The interplay of all these determinants during gravitropic root bending, together with consequences of additional effectors of root growth on such crosstalk (e.g. brassinolide), however, is far from being understood. Therefore, we carefully edited our manuscript, emphasizing the point that conclusions drawn from our experiments and simulations apply to the conditions tested, with all the limitations arising from the fact that additional parameters remain to be addressed in more detail in further experimental setups.

The authors support for their interpretation of PIN2 asymmetry diminishing auxin asymmetry currently relies solely on their computational model. Unfortunately, there are several serious problems with this model, precluding that we currently accept its outcomes at face value.

The first and foremost problem lies in the interplay of what they investigate -PIN2 asymmetry- with the boundary condition of connecting to a virtual shoot. Of course, a connection to a virtual shoot with influx coming from this shoot into the vasculature and efflux from the outer tissues to the shoot should be incorporated in these types of models. However, in the current model, it is this virtual shoot connection that is probably generating the results. Put simply, if you make the virtual shoot connection directly next to the elongation zone in which the PIN2 asymmetry is applied you simply get that more PIN2 causes more export to the virtual shoot and hence less auxin, and hence a reversal between PIN2 and auxin asymmetry when PIN2 asymmetry is simulated in isolation and the repressive effect of PIN2 asymmetry on auxin asymmetry when simulating gravitropism. This is in my opinion an artifact of the model implementation, not a genuine result!

Response: Thank you for pointing this out, as it signifies some shortcomings in the presentation of our model in the original manuscript. In the revised version we included different simulations, in which we tested different geometries and boundary conditions. This includes simulations, in which we tested different distances between the root elongation zone and the 'source' and 'sink' boundaries. Apart from that, we included simulations with different patterns of PIN2 expression/activities (throughout the entire root vs. elongation and cell division zone) as well as different PIN2 levels (see Fig. S11, Fig. S12, Fig. S11g and Fig. S12d for model comparisons, Fig. S14a for PIN2 level effects). Importantly, in all tested simulations we observed an inverse correlation between auxin levels and PIN2 abundance (in Fig. S11g, Fig. S12d, Fig. S14a, the slope of the graph is always positive, as in the main figure Fig. 8g).

It is in this context important to note that in the model the auxin asymmetry in the lateral root cap, which has no shoot connection and hence not this auxin loss, is aligned with the PIN2 asymmetry: more auxin where there is more PIN2 (Fig 8b)!

Response: this impression might have been caused by the poor representation of our simulation results: According to our simulations there is less auxin in LRC at the upper side, when there is more PIN2 at the plasma membrane. This is in agreement with the outcome of our PIN2 simulations in epidermis and cortex cells. To avoid further misunderstandings and to improve visibility, we redraw our model geometry with wider LRC cells (this change does not affect the outcome of our simulations), in our revised Fig. 8.

The severity of the auxin loss and its interference with the PIN2 asymmetry under investigation can be easily avoided by incorporating also a differentiation zone in the model, in which PIN2 levels taper off and PIN patterns become more lateralized -thereby significantly reducing severity of auxin loss- and also no PIN2 asymmetry occurs under gravitropism (see eg. Figure 1 in Baster et al, 2013 *Embo Journal*) -thus separating the boundary

condition from the subject of study- and then have the virtual shoot connection attached to this more shootward differentiation zone.

Response: We thank the reviewer for this suggestion, and agree that it is very important to emphasize the effects of boundary conditions on the outcome of our model.

Our initial geometry is 1200 μm long and contains $\sim 500 \mu\text{m}$ of differentiation zone (see Fig. S11a). In the revised manuscript we have added a 3500 μm -long root geometry (see Fig. S11b), with 2700 μm simulating the differentiated portion of the root apical of the root elongation zone. As pointed out by reviewer #3 we found that variations in the distances between 'source' and 'sink' influence the absolute value of auxin concentrations in the root apex. However, we still observed an inverse correlation between auxin ratios (IAA_{lower/upper} side) and PIN2 ratios (PIN2_{upper/lower} side), which is summarized in Fig. S11g.

As proposed by reviewer 3, we have also tested the effect of lateral PIN2 in the epidermal layer and cortex (Fig. S13a) and indeed our simulation shows that such lateralization of PIN2 reduces auxin loss by returning auxin into the stele. This causes increased absolute values of auxin concentration in the root apex compared to the differentiation zone (Fig. S13a, right part). However, an inverse correlation between auxin ratios (IAA_{lower/upper} side) and PIN2 ratios (PIN2_{upper/lower} side), when introducing PIN2 asymmetry still holds (Fig. S13e). This result indicates that this inverse correlation does not depend on boundary conditions and on overall auxin amounts in the root tip.

The auxin loss problem is further aggravated by the specific model settings used by the authors, who in contrast to other recent studies do not incorporate auxin induced AUX1 expression that can help hold on to auxin if there is more auxin, and also do not incorporate laterally inward oriented PIN2 in the epidermis and cortex enabling reflux of auxin into the vasculature. As a consequence in the model used here there will be a fairly strong positive correlation between auxin loss and PIN2 level.

Response: We fully agree with reviewer #3. AUX1 needs to be considered for any modeling that simulates auxin distribution in gravistimulated roots. Whilst, we did not observe any obvious responses of AUX1-YFP expression pattern in response to eBL (Fig. S8), it cannot be excluded that AUX1 activity is modulated independently of overall protein abundance. This might happen upon gravitropic root bending, for example, as a consequence of localized pH adjustments in such roots. Dependence of AUX1 affinity to auxin on pH was investigated e.g. by Carrier et al. (Plant Physiol.; 2008 doi: [10.1104/pp.108.122044](https://doi.org/10.1104/pp.108.122044)). However, we were not able to include such a complicated dependence of AUX1 permeability on auxin concentrations into our current model. This is mainly due to the lack of data, concerning the exact pH values during gravitropic response, and its dependence on auxin concentration.

A relevant parameter that could concern the outcome of our model, relates to hypothetical overall changes in AUX1 activity in eBL-treated roots. We therefore investigated a wide range of AUX1 permeabilities (Fig. S15) and its effects on our simulation. With such a wide range of hypothetical permeabilities we tried to account for different mechanisms causing either induction or downregulation of AUX1 activity. Again, an inverse correlation between auxin ratios (IAA_{lower/upper} side) and PIN2 ratios (PIN2_{upper/lower} side) holds true for any value of AUX1 (Fig. S15b), which is in agreement with the outcome of our main model presented in Fig. 8, indicating that a lateral PIN2 gradient causes dampening of an established auxin gradient.

In a separate model setup, we investigated effects of PIN2 located to lateral domains of epidermis and cortical cells (Fig. S13a) and quantified consequences on auxin ratio in dependence of a PIN2 gradient formed in gravity-responding roots (Fig. S13e). The outcome of this simulation demonstrates that variations in PIN2 subcellular localization impact on the absolute values of auxin concentrations (lateral PIN2 causes a more pronounced accumulation of auxin in the root tip). However, no effects on the auxin ratios in dependence of the PIN2 gradient could be observed, even if lateral PIN2 permeability also becomes asymmetrical to the same extent as apical PIN2.

I strongly suspect that if the authors include an extended differentiation zone with tapering off, symmetric PIN2 levels, laterally oriented PIN2, and auxin dependence of AUX1 expression that were both recently shown to be critical for root tropisms, auxin and PIN2 asymmetry will also be aligned in the elongation zone and PIN2 asymmetry will contribute to rather than diminish auxin asymmetry. Notably, this would require an alternative explanation for why eBL enhances gravitropic responses while reducing PIN2 asymmetry.

Response: Thanks for pointing this out! We do believe that we have commented on all of these points in the sections above. In short, we tested different root geometries, boundary conditions, distances from EZ to the "source" and "sink" boundary, different patterns of PIN2 activities and different PIN2 and AUX1 levels (see Fig. S11, S12, S13, S14, S15), and in all cases our modeling indicated a relative increase in auxin levels in cell files with reduced PIN2 levels, and *vice versa*. This is in agreement with the conclusion that is based on the outcome of our modeling; i.e. the ratio of IAA_{lower/upper} concentrations is proportional to the ratio of PIN2 found at the upper vs. lower side of gravistimulated roots.

For all these setups, IAA_{lower/upper} can be predicted by considering the simplest case, with just one cell having influx and efflux carriers, in which auxin concentration is inversely proportional to PIN2 concentration (see also our simplified model in Fig. S12).

Note also that the squarish root topology used by the authors has been abandoned by most research groups in the field in favor of a realistic root tip layout, and that this has been demonstrated to impact the auxin patterns obtained for simulated tropisms.

Response: We fully agree with this point! Recent published work made use of a realistic root tip layout instead of the 'squarish' representation of root meristems. We therefore tested our simulation on an image-based root geometry and found no qualitative differences from 'squarish' geometry (Fig. S11d,g). The realistic root tip layout is essential for simulating absolute auxin concentration values and for relative concentrations in different cell layers and cell types. In contrast, here we calculate auxin ratios in cells which are symmetrically situated on two sides of the root meristem ("mirror cells"), and it appears that for this ratio, geometry of the rest of the root is less relevant, than for 'true concentration value' simulations. However, we found that in the realistic root tip layout, auxin ratio establishment in dependence of asymmetric PIN3/7 distribution, differs considerably from the squarish root geometry (Fig. S11g). This supports current views, indicating that auxin fluxes from columella cells to the elongation zone depend very much on cell shapes. This difference in auxin flux, when comparing squarish and realistic geometries, however does not interfere with the inverse proportionality of auxin concentrations at the upper and lower side of gravistimulated roots and the abundance of PIN2 in those cells.

Finally, it seems that the authors apply a single polar orientation for cortical PIN2 yet data show a flip in cortical PIN2 polarity from meristem to elongation zone. Again, this has been shown to be important for gravitropism.

Response: We agree with reviewer #3. We missed this point in the original version of our manuscript. Therefore, by employing a separate model setup, we have investigated the effect of basal cortical PIN2. Results of these simulations are summarized in Fig. S13b-d. In the main model, basal cortical PIN2 permeability was always unchanged and equal to the initial PIN2 permeability. In the new setup we calculated auxin ratios under conditions when PIN2 asymmetry is established, and included basal PIN2 permeability in cortical cells (Fig. 13c). In addition, we tested a hypothetical scenario, in which there is no PIN2 present in the root cortex (Fig. S13d). Such variations in cortical PIN2 localization affected absolute auxin concentrations (Fig. S13d), but did not affect the auxin ratio that we first observed in our main model. The outcome of these simulations thus appears consistent with the outcome of our main model, indicating an inverse correlation between auxin ratios (IAA_{lower/upper} side) and PIN2 ratios (PIN2_{upper/lower} side) (Fig. S13e).

In conclusion, I can not recommend the paper in its current form for publication and recommend that particularly the modeling needs to be substantially improved.

Reviewer #4 (Remarks to the Author):

The hormonal regulation of plant growth and development represents a major focus by Plant Science researchers. The Arabidopsis root has provided a popular experimental system where many of the most important mechanistic insights have arisen. The current manuscript by Retzer et al, reports how another important class of plant hormone termed brassinosteroids (BR) regulates gravitropism via the auxin efflux carrier PIN2. This represents a significant finding as this important root adaptive response is reported to be primarily controlled via auxin.

Response: Thank you very much for these encouraging lines!

The manuscript initially describes how the authors screened for signals controlling endocytosis of ubiquitinated PIN2 (a focus of the lab) leading to the discovery that BR plays a key role. Through a series of elegant experiments using novel tools, the authors convincingly demonstrated that BR caused PIN2 stabilisation at the plasma membrane (PM). By monitoring a YFP tagged ubiquitinated PIN2 form in the background of several BR synthesis and signalling mutants or equivalent chemical inhibitors, the authors observed that the BR transduction pathway controls trafficking of this key root auxin transport protein. They went on to demonstrate that this does not reflect BR regulated changes in PIN2 gene expression, but that BR impacts this ubiquitinated protein's endocytotic sorting via the TOL/ESCRT machinery.

As I read and digested the significance of the results described in the manuscript, I wanted to understand how this may impact the role of PIN2 during root gravitropism, as a clear function for BR has not been established with this adaptive response to date. I was therefore delighted that the authors next considered the wider adaptive importance of this BR-auxin crosstalk mechanism by studying the impact of BR treatment and/or mutants on root gravitropic bending kinetics. Intriguingly, they observed that BR treatment conferred a hyper-responsive behaviour, as wild type (WT) roots (but not *pin2*) exhibited over-bending after a 90 degree gravity stimulus.

To determine which phase(s) of the root gravitropic response BR regulation of PIN2 may be important, the authors characterised the behaviour of PIN2-VENUS reporter* during a root gravitropic response. They observed that PIN2-VENUS undergoes elevated levels on the lower side of gravity stimulated roots (after 90'), which was disrupted in BR treated roots.

*Did they also monitor the effect of gravistimulation on the PIN2-Ub-VENUS reporter?

Response: In the revised version of our manuscript, we included an analysis of *eir1-4 PIN2p::PIN2:ubq:VEN* and tested its gravitropic root bending in the presence of brassinolide (Fig. S9b). We observed a partial restoration of gravitropic root growth under these conditions, indicating that the PIN2::ubq:VEN fusion protein has retained some functionality, which substantiates the outcome of our various experiments performed with this reporter line. Furthermore, partial restoration of root gravitropism in eBL-treated *eir1-4 PIN2p::PIN2:ubq:VEN* is in line with our working hypothesis, in which eBL-induced stabilization of PIN2 at the plasma membrane would be sufficient for differential auxin transport into the root elongation zone.

To understand the impact that this transient up-regulation of PIN2 on the lower root side may have on auxin levels, the authors employed a previously published mathematical model. The model revealed that this regulatory mechanism served to counter-act the earlier formation (within minutes) of the lateral auxin gradient. Hence, the BR-PIN2 mechanism appears to function during the last phase of a gravitropic response, serving to attenuate the lateral auxin gradient, stopping the root over-bending. This point could be spelt out much clearer in the discussion (which currently under sells the significance of the authors findings). To date, little is known about this last phase of a gravitropic response other than gravity-sensing statolith's rolling back into position as a root reaches its mid-point in bending.

Response: We really appreciate this very positive response and encouragement! Thank you very much indeed! As you will see, the revised version of our m/s now includes additional modeling datasets in which we included simulation of determinants of gravitropic root bending, other than PIN2. Simulations and experiments support the suggested role for PIN2 gradient formation, dampening auxin gradient formation rather than enhancing it, under all conditions tested/simulated. Furthermore, we considered additional parameters of auxin transport in gravity-responding roots (*AUX1*, etc.) in our simulations, which highlighted their contribution to differential auxin transport and root bending. Understanding the interplay of these different determinants of gravitropic root bending during early and later stages of gravitropic root bending, still remains a challenge for future research. Perhaps, the algorithms developed for our extensive simulations of auxin distribution in gravity-responding root tips, will be helpful for such follow-up research.

I was left wondering after reading the manuscript, so where does BR signalling fit into a root gravitropic response? It's clear from the authors use of the BR response reporter BES1-GFP that there is a transient BR response on the lower side of roots 90' after a gravity response, coinciding with elevated PIN2 levels, but how is the BR response induced? In fairness, this is beyond the scope of the current manuscript, but it's a question I hope the authors or readers of this manuscript will follow up in the future.

Response: This is an important question, and quite frankly, we have -so far- no good idea, as to how such differential brassinolide signaling might be induced in the first place. In the revised version of the m/s, we tested consequences of externally applied brassinolide on BES1-GFP gradient formation in gravistimulated roots. Such treatment antagonized gradient formation and thus very likely differential brassinolide signaling. Modifications in PIN2 sorting and gravitropic root bending that we observed under these conditions thus could -at least partially- result from interference with differential brassinolide signaling. Apart from that, brassinolide has been demonstrated to impact on a wide range of additional determinants of root growth/cell differentiation affecting root twisting/curling. From that it appears that the apparent brassinolide effect on PIN2 is very likely not the only determinant, mediating crosstalk between brassinolide signaling and root gravitropism. Meanwhile, we started introducing the BES1-GFP into various lines/mutants, compromised in diverse signaling pathways. Analysis of the resulting material should shed some light onto this question.

Reviewers' comments:

Reviewer #1 (Remarks to the Author):

The authors substantially improved their ms and addressed most of my concerns. I have only two comments:

1. I think the authors should also explain the reader why pin212K-R and pin217K-R lines were not used in this work while eir1-4 PIN2p::PIN2:ubq:VEN is valid. This is a useful information for the community when designing follow up studies.
2. Regarding the wild type version of PIN2-VEN, my previous point 11 remained somewhat unsolved. The new image in Fig 6 does not show the comparison between 0 and 150. In any case, the level of PIN2 in the upper side membrane is enhanced in response to BL, as inherently expected from the ratio measurements. I therefore wonder why this PIN2 baseline signal in the PM is not enhanced by BL, in images as in Fig. 4d e.

note: Supplemental, Figure S1 legend: panel J?

Reviewer #2 (Remarks to the Author):

The authors have provided additional experimental data showing that the pertinent constructs used indeed have at least partial functionality. Because that was my main concern, I find the revised manuscript very much improved and like to commend the authors with this very nice story.

Sacco de Vries

Reviewer #3 (Remarks to the Author):

Let me start by complimenting the authors with the thoroughness with which they addressed my concerns. I have only a few remaining minor issues.

Minor points:

1. It would help the reader if in figure 8h it were clearly indicated that all these simulations are done for $k=1$. Additionally, I think it is nice to mention that the model reproduces the pin2 agravitropism phenotype: very low PIN2 permeability results in absence of auxin asymmetry even under very strong PIN3/PIN7 asymmetry. This is an important benchmark for the validity of the model that deserves underlining.
2. What is the experimentally observed increase in PIN2 membrane levels under eBL treatment? It would help to discuss this in relation to figure 8h to help readers judge whether e.g. the 2 or 5-fold increases are already unrealistically high or not.
3. Why not do figure 8h for $k=0.81$, eBL, by inducing a $k=1$ already has a 1.93 auxin asymmetry as opposed to the 1.62 control asymmetry, leaving much less additional asymmetry to explain by changes in PIN2 levels?
3. Supplementary figure 11: where can I see the effect of having / not having the same AUX1/PIN2 pattern in the differentiation zone cells incorporated in the elongated root geometry model?

Reviewer #4 (Remarks to the Author):

The authors have provided a comprehensive response to all 4 reviewers comments and, in my view, addressed the key issues raised by myself and other reviewers in the revised manuscript.

One minor point - the last results section has expanded considerably with the inclusion of many details about the modelling activities. I recommend this information is replaced with a much shorter summary and the model details are put into an SI section which is referenced in the text.

Response to reviewers

Reviewer #1 (Remarks to the Author):

The authors substantially improved their ms and addressed most of my concerns. I have only two comments:

Response: thank you very much for this positive feedback, and all the extremely helpful comments that we tried to implement into our manuscript!

1. I think the authors should also explain the reader why pin212K-R and pin217K-R lines were not used in this work while eir1-4 PIN2p::PIN2:ubq:VEN is valid. This is a useful information for the community when designing follow up studies.

Response: we fully agree with this point. We have added this information to the Methods section of the revised m/s. Now, this relevant information can be found in the manuscript correspondence and in the manuscript file itself.

2. Regarding the wild type version of PIN2-VEN, my previous point 11 remained somewhat unsolved. The new image in Fig 6 does not show the comparison between 0 and 150. In any case, the level of PIN2 in the upper side membrane is enhanced in response to BL, as inherently expected from the ratio measurements. I therefore wonder why this PIN2 baseline signal in the PM is not enhanced by BL, in images as in Fig. 4d e.

Response: In the revised version of Fig. 6, we now have added two more images showing signal intensities/distribution at time point zero (Fig. 6a,b). As pointed out by reviewer #3, there is an inherent increase in PIN2-Venus signal intensities at the upper side of gravistimulated roots in response to eBL. We quantified this difference by determining the PM/intracellular signal ratio in control and in eBL-treated roots (Fig. S15c). We found that eBL causes increased plasma membrane signal intensities only at the upper side (when compared to controls), whereas no pronounced difference was detectable at the root's lower side. From that it appears that eBL stabilizes PIN2 only at the upper side of the root. This supports our model, in which eBL acts as a signal that specifically antagonizes enhanced PIN2 endocytic sorting and degradation, induced -for example- by gravistimulation. We tried to stress this observation in our m/s.

We can only speculate about the limited eBL effects that we observed in the absence of signals/stimuli inducing degradation of PIN2 (compare, e.g. Fig. 4d,e 'stable' conditions and Fig. 4 g-l; Fig. 6a-d – 'destabilized' conditions). Perhaps, differences in intracellular sorting of 'stable' PIN2 and PIN2 bound for degradation contribute to these distinct responses. In addition, we cannot exclude that additional factors and determinants contribute to the stabilization of PIN2 in response to eBL. For example, the cascade of events associated with PIN2 (poly)ubiquitylation followed by ESCRT-

dependent sorting into late endosomes, might include some regulatory switches that are controlled by variations in brassinolide signaling. This might specifically concern ubiquitylated PIN2, whereas non-ubiquitylated PIN2 would remain unaffected. Such a scenario would explain that PIN2:ubq:VEN exhibits more pronounced eBL responses, than wild type PIN2:VEN. Clearly, future experimentation will be required to provide a definite answer to this question.

note: Supplemental, Figure S1 legend: panel J?

Response: Thanks for spotting this! We have changed the Figure accordingly.

Reviewer #2 (Remarks to the Author):

The authors have provided additional experimental data showing that the pertinent constructs used indeed have at least partial functionality. Because that was my main concern, I find the revised manuscript very much improved and like to commend the authors with this very nice story.

Sacco de Vries

Response: Thank you very much for this positive feedback!

Reviewer #3 (Remarks to the Author):

Let me start by complimenting the authors with the thoroughness with which they addressed my concerns. I have only a few remaining minor issues.

Response: Thank you very much indeed, for all your feedback, which helped us to substantially improve our manuscript!

Minor points:

1. It would help the reader if in figure 8h it were clearly indicated that all these simulations are done for $k=1$. Additionally, I think it is nice to mention that the model reproduces the pin2 agravitropism phenotype: very low PIN2 permeability results in absence of auxin asymmetry even under very strong PIN3/PIN7 asymmetry. This is an important benchmark for the validity of the model that deserves underlining.

Response: we have added this information in the revised version of Fig. 8h and we edited our main manuscript, highlighting the relationship between PIN3/PIN7 asymmetry and PIN2 permeability (lines 341-343).

2. What is the experimentally observed increase in PIN2 membrane levels under eBL treatment? It would help to discuss this in relation to figure 8h to help readers judge whether e.g. the 2 or 5-fold increases are already unrealistically high or not.

Response: Thank you very much for pointing this out! We now refer to this point in our revised m/s (lines 354-356), and added the experimental data (Fig. S15c). Since overall PIN2 levels do not change dramatically in response to eBL (judged from Western blot analyses; Fig.4d-f, Fig. S7gh), we determined the plasma membrane to intracellular signal ratio in gravistimulated control and eBL-treated roots. The PM/intracellular ratio of PIN2-Venus signals at the root's upper side increased by 1.83 in response to eBL treatment. At the lower side, there was essentially no increase detectable in response to eBL ($= 1.1$). Thus, upon gravistimulation, brassinolide seems to promote PIN2 plasma membrane retention, specifically at the upper side of the root. These values appear at the lower margin, when compared to the minimum 2- to 5-fold increases in permeability values that we obtained in our simulations.

3. Why not do figure 8h for $k=0.81$, eBL, by inducing a $k=1$ already has a 1.93 auxin asymmetry as opposed to the 1.62 control asymmetry, leaving much less additional asymmetry to explain by changes in PIN2 levels?

Response: Another relevant point that we implemented in our revised m/s. We added the requested graph as part of our Supplementary Information (Fig. S15a,b). Indeed, when comparing curves for $k=0.81$ and $k=1$, it turns out that a smaller increase in PIN2 levels is required for the observed auxin asymmetry in eBL-treated samples, which is within the range of our experimental data (Fig.15c).

3. Supplementary figure 11: where can I see the effect of having / not having the same AUX1/PIN2 pattern in the differentiation zone cells incorporated in the elongated root geometry model?

Response: Thank you very much for pointing this out! We added the requested information to our revised manuscript - Figure S11d.

Reviewer #4 (Remarks to the Author):

The authors have provided a comprehensive response to all 4 reviewers comments and, in my view, addressed the key issues raised by myself and other reviewers in the revised manuscript.

Response: Thank you very much!

One minor point - the last results section has expanded considerably with the inclusion of many details about the modelling activities. I recommend this information

is replaced with a much shorter summary and the model details are put into an SI section which is referenced in the text.

Response: We agree with reviewer #4 – we put quite some detailed information about modelling and its various effects on auxin ratios. In the revised version of the manuscript we tried to find a compromise: leaving the modelling and its conclusions in the main text, whereas sections describing model construction have been moved to Supplementary Information.

REVIEWERS' COMMENTS:

Reviewer #1 (Remarks to the Author):

The authors addressed all my concerns, thank you.

Reviewer #3 (Remarks to the Author):

The authors have done a great job amending all comments and I recommend the article now for publication now